# EVADING DATA CONTAMINATION DETECTION FOR LANGUAGE MODELS IS (TOO) EASY

## ABSTRACT

The benchmark performance of large language models (LLMs) has a high impact on their popularity and is thus of great importance to many model providers. However, the reliability of such benchmark scores as a measure of model quality gets compromised if the model is contaminated with benchmark data. While recent contamination detection methods try to address this issue, they overlook the possibility of deliberate contamination by malicious model providers aiming to evade detection. We propose a categorization of model providers based on their (de)contamination practices and argue that malicious contamination is of crucial importance as it casts doubt on the reliability of public benchmarks. To study this issue more rigorously, we analyze current contamination detection methods based on their assumptions. This analysis reveals a significant vulnerability in existing approaches: they do not account for rephrased benchmark data used during training by malicious actors. We demonstrate how exploiting this gap can result in significantly inflated benchmark scores while completely evading current detection methods. [1]

## 1 INTRODUCTION

The recent popularity of large language models (LLMs) and their applicability to a wide range of tasks has led to significant investments in the field, with many companies competing to train the best model (Anil et al., 2023a; Jiang et al., 2023; OpenAI, 2023; Touvron et al., 2023). Accurately assessing the quality of these models is thus not only crucial to tracking progress in the field and choosing the correct model for a specific task but also has significant economic implications. As a result, many high-quality benchmarks have been developed for a wide range of tasks (Clark et al., 2018; Cobbe et al., 2021; Hendrycks et al., 2021; Lin et al., 2022).

**Contamination Detection** These benchmarks are typically made public to allow evaluation of new models. However, as LLMs are often trained on scraped web data, benchmark samples may inadvertently become part of the training dataset. This *data contamination* can lead to inflated benchmark performance and inaccurate evaluation results. To alleviate this issue, both model providers (Anil et al., 2023a; OpenAI, 2023; Touvron et al., 2023) and third parties (Golchin and Surdeanu, 2023a; Oren et al., 2023; Shi et al., 2023) developed methods to detect and quantify the influence of data contamination on model performance.

**Malicious Actors** However, high competitive pressure and significant financial stakes could incentivize malicious actors to *actively contaminate* their model to increase benchmark performance while *evading detection*. Crucially, current contamination detection methods do not account for such malicious behavior, which presents a significant oversight.

**This Work: Evading Detection** We demonstrate that *all current detection methods can be evaded* by training on rephrased benchmark samples (see Fig. 2) while still boosting performance significantly. This endangers the integrity of current benchmarks and highlights the need for a systematic study of contamination detection in the malicious setting. Furthermore, the simplicity of this approach indicates that malicious actors might already be exploiting this gap in real-world scenarios.

---

[1]Code is available in the supplementary material.

Figure 1: Overview of four archetypes for model training. Malicious, honest-but-negligent, and proactive actors perform different data preprocessing. Evasively malicious actors perform additional steps to avoid contamination detection. This allows the malicious actor to get the best clean performance. Attribution in App. A.

**Systematizing (De-)Contamination Practices**
To facilitate a rigorous analysis of contamination detection and evasion methods, we first define four model provider archetypes, depending on their (de-)contamination practices. We illustrate the whole training and evaluation pipeline for each archetype in Fig. 1: *proactive actors* take active measures to decontaminate their training data effectively, *honest-but-negligent actors* do not actively contaminate their training data but take no or ineffective precautions to prevent contamination, and *malicious actors* actively contaminate their training data to increase benchmark performance. Among malicious actors, we further differentiate between *openly malicious* and *evasively malicious* ac-

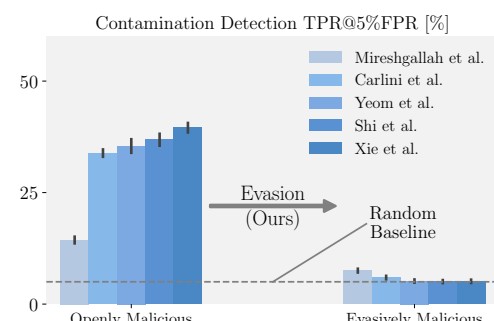

Figure 2: Evading contamination detection can be done very effectively. We show the 2-sigma intervals for the reported bars in the plot. TPR, resp. FPR, refers to the true, resp. false, positive rate.

tors, with the latter taking extra steps to evade detection. We review current decontamination practices with respect to these categories and conclude that most model providers are likely honest-but-negligent (Almazrouei et al., 2023; Anil et al., 2023a; Jiang et al., 2023; OpenAI, 2023; Touvron et al., 2023), casting doubt on their model's performance.

**Evading Contamination Detection**    Finally, we review current detection methods w.r.t. the assumptions they (implicitly) make about the model provider and model access. This analysis reveals a critical oversight in handling rephrased data, allowing us to propose a technique that rephrases benchmark samples during finetuning and targeting detection methods with and without access to the training data. We show that this simple technique can evade all current detection methods (see Fig. 2) and significantly improves benchmark performance by up to 15%.

**Key Contributions**    Our key contributions are:

- We define four (de-)contamination settings, highlighting the risks of malicious actors (§3).
- We discuss the assumptions made by current contamination detection methods (§4) and reveal a critical oversight that enables us to propose a simple yet effective rephrasing-based detection evasion technique (§5).
- We demonstrate that our attack evades all current detection methods while still significantly improving benchmark performance by up to 15% (§6).

## 2    DATA CONTAMINATION

Before systematizing (de-)contamination practices and detection methods, we first need to formally define data contamination. We consider a (training) dataset $\mathcal{D}$ to be contaminated with some benchmark $\mathcal{D}'$ if there is an overlap between the two. We call models trained on a contaminated dataset *contaminated models*.

From a detection perspective, it is helpful to differentiate between *sample-* and *benchmark-level* data contamination. Sample-level contamination detection aims to determine whether a given sample $x$ was contained in the training data $\mathcal{D}$. In contrast, benchmark-level detection aims to determine whether any subset of a benchmark $\mathcal{D}'$ was contained in $\mathcal{D}$ without specifically aiming to provide this subset. More formally, we define sample- and benchmark-level contamination as follows:

**Definition 1** (Sample-level Data Contamination). *A dataset $\mathcal{D}$ is contaminated with a sample $x$ from a benchmark $\mathcal{D}'$ if $x \in \mathcal{D}$.*

**Definition 2** (Benchmark-level Data Contamination). *A dataset $\mathcal{D}$ is contaminated with a benchmark $\mathcal{D}'$ if $\mathcal{D}' \cap \mathcal{D} \neq \emptyset$.*

*Sample-level* detection methods provide fine-grained information on the amount of contamination in a dataset. By excluding the detected samples, model providers can present evaluation results on a clean subset of the benchmark (Brown et al., 2020; Touvron et al., 2023). However, as detection errors can significantly influence evaluation results and partial contamination can impact performance on the uncontaminated benchmark portion, it is questionable whether these results are comparable to an uncontaminated model.

*Benchmark-level* contamination is particularly relevant to a benchmark's integrity as a performance metric. If a model has been contaminated with a benchmark, its results are not comparable to those of an uncontaminated model. However, benchmark-level methods do not specify which samples were contaminated, making it challenging to assess the contamination's effect on model performance.

The use of a contaminated dataset for model training does not necessarily have an influence on the model's measured performance. From an adversarial perspective, this form of contamination is highly uninteresting. We therefore define *problematic* data contamination as data contamination that leads to an increase in performance on a benchmark. More formally, we define:

**Definition 3** (Problematic Data Contamination). *A dataset $\mathcal{D}$ is problematically contaminated with a benchmark $\mathcal{D}'$ if $\mathcal{D}$ is contaminated with $\mathcal{D}'$ according to Definition 2 and a model trained on $\mathcal{D}$ obtains a higher performance on $\mathcal{D}'$ than a model trained using the same training method on $\mathcal{D} \setminus \mathcal{D}'$.*

For the rest of this paper, we focus on problematic data contamination.

## 3 MODEL PROVIDERS AND DATA CONTAMINATION

Considering the significant effect benchmark contamination can have on a model's measured performance (Brown et al., 2020; Sainz et al., 2023; Touvron et al., 2023; Yang et al., 2023) and the high stakes involved in training LLMs, there are strong incentives for model providers to not fully decontaminate their models. Therefore, it is essential to consider the possibility of negligent or malicious behavior when studying contamination detection.

To facilitate a more nuanced study, we define four model provider archetypes, differentiating between active contamination (either open or covert), active decontamination, and honest-but-negligent indifference.

### 3.1 ACTOR ARCHETYPES

We distinguish model providers or actors based on the actions they take (or neglect to take) to prevent (or cause) data contamination, leading to the following definitions:

**Definition 4** (Actor Roles). *A malicious actor actively contaminates a model by deliberately using benchmark data during training with the goal of artificially increasing benchmark performance.*

*An* honest-but-negligent *actor possibly contaminates a model by not taking sufficient measures to prevent data contamination but does not actively contaminate the model.*

*A* proactive *actor actively decontaminates a model by taking sufficient measures to guard against contamination, ensuring representative performance on a benchmark.*

The line between these types can be blurry. For instance, an actor taking reasonable but incomplete decontamination measures can fall between proactive and honest-but-negligent. However, a more fine-grained distinction is not necessary in the context of this paper.

**Evasiveness** As a malicious actor might try to evade contamination detection, it is crucial to distinguish between openly malicious and evasively malicious actors:

**Definition 5** (Evasiveness). *We distinguish between* openly malicious *and* evasively malicious *actors depending on whether they actively try to hide the use of benchmark data by modifying the training or data preprocessing procedure with the goal of evading contamination detection.*

This distinction is particularly important when evaluating contamination detection methods as prior works (Carlini et al., 2021; Mireshghallah et al., 2022; Shi et al., 2023; Yeom et al., 2018) fail completely in the evasively malicious setting (see Fig. 2). In real-world scenarios, a malicious actor is likely to use evasion strategies to avoid detection. Unfortunately, the current literature does not consider this possibility and solely focuses on openly malicious actors, making their applicability in real-world scenarios questionable. Before discussing these detection methods, we review current data decontamination practices among model providers.

## 3.2 Current Decontamination Practices

As prior work (Li and Flanigan, 2023; Sainz et al., 2023) found indications of widespread data contamination in popular models, we review the decontamination practices reported in the corresponding publications. Specifically, Sainz et al. (2023) found that common training datasets are heavily contaminated with current benchmarks and Li and Flanigan (2023) found that most models perform significantly better on benchmarks released before the model.

Most model and dataset providers do not describe any active decontamination measures (Abdin et al., 2024; Almazrouei et al., 2023; Anil et al., 2023a;b; Anthropic, 2024a;b; Chowdhery et al., 2022; Computer, 2023; Dubey et al., 2024; Jiang et al., 2023; Mehta et al., 2024; OpenAI, 2023; Penedo et al., 2023; Young et al., 2024), likely placing them in the honest-but-negligent category. However, several providers do report deduplication protocols (Anil et al., 2023b; Brown et al., 2020; Computer, 2023; Penedo et al., 2023) which are believed to increase performance (Lee et al., 2022).

A post-hoc contamination analysis, which involves evaluating models only on the uncontaminated portion of a benchmark, is more common (Anil et al., 2023a; Brown et al., 2020; Dubey et al., 2024; OpenAI, 2023). However, as we will show in §6, this is insufficient, since partial contamination can still significantly improve performance on the uncontaminated portion of the benchmark (see e.g. Table 1). Furthermore, this post-hoc analysis is typically not reproducible as neither training datasets nor indices of the evaluated test set portions are made available (Anil et al., 2023a; OpenAI, 2023; Dubey et al., 2024). Combined, this makes it exceedingly difficult to meaningfully compare models even among honest-but-negligent actors.

While *proactive decontamination is essential to ensure fair and meaningful model comparisons*, we only found descriptions of such measures in Brown et al. (2020); Chowdhery et al. (2022); Yang et al. (2024) among the works we reviewed. Brown et al. (2020); Yang et al. (2024) perform the overlap check a-priori, removing all benchmark samples from the training set. Chowdhery et al. (2022) only describes filtering for a canary string included in the BigBench benchmark (Srivastava et al., 2022), which is a unique string that should be in all documents containing the dataset.

## 4 Detecting Data Contamination

While sample-level contamination detection is well-studied under the name membership inference attack (MIA) in privacy research (Carlini et al., 2021; Shokri et al., 2017; Song and Shmatikov, 2019), benchmark-level detection has only been investigated recently (Golchin and Surdeanu, 2023a; Oren et al., 2023). Interestingly, transferring methods from the sample- to the benchmark-level setting is not trivial, since false positives can result in noisy signals.

To facilitate a rigorous discussion of contamination detection methods, we review current methods based on their assumptions and introduce several dimensions for categorization, referring to App. B for a full overview. We will leverage this analysis to identify a critical issue that can be easily exploited by malicious actors in §5.

### 4.1 Detector Assumptions

**Access** MIAs typically consider three levels of access to the model: black-box, grey-box, and white-box. *Black-box access* allows access to the model's predictions only, *grey-box access* also

includes the predicted confidences, and *white-box access* provides full access to model weights and parameters. In the context of data contamination, some methods additionally require access to all training data. We call this fourth level *oracle access*. While traditional MIAs become trivial in this setting, the huge training sets of LLMs make detection challenging even with this level of access.

Black-box methods often work by comparing the model's performance on a benchmark to the performance on other data (Huang et al., 2023; Zhu et al., 2023) or check for verbatim memorization of sample text (Golchin and Surdeanu, 2023a). Black-box methods are especially relevant for models that are only available through an API (Anil et al., 2023a; OpenAI, 2023).

Grey-box methods (Mattern et al., 2023; Shi et al., 2023) leverage the model's perplexity or certainty on a given sample and often perform better than black-box methods. They are applicable to models with more extensive API access and all open-weight models. We are not aware of any white-box access methods, although they would also be applicable to all open-weight models.

Oracle access methods (Yang et al., 2023) are the most powerful but can only be used by the model providers themselves, as training data is generally not published. These methods are based on similarity checks between training and benchmark data and can be used to check for contamination before or after model training (OpenAI, 2023; Touvron et al., 2023).

**Metadata** Metadata refers to all information related to a benchmark that is not part of the actual samples. This includes the dataset name (Golchin and Surdeanu, 2023a;b) and a canonical ordering of samples (Oren et al., 2023). If such metadata has been learned, this is a strong indication of contamination. However, metadata contamination is a strong assumption. Not only can malicious model providers simply remove the metadata, but benign dataset shuffling will remove the canonical ordering of samples and limited context length makes the association of dataset names with individual samples unlikely.

**Reference Models** Many methods require access to uncontaminated reference models (Mireshghallah et al., 2022; Song and Shmatikov, 2019; Watson et al., 2022). However, Mattern et al. (2023) shows that reference-based methods are highly sensitive to the reference model. Furthermore, obtaining uncontaminated but comparable reference models is often not feasible.

Several methods do not explicitly require a reference model but do require a threshold on a contamination score to decide when a sample should be considered contaminated (Carlini et al., 2021; Li, 2023a; Shi et al., 2023). This makes it difficult to apply these methods without a reference model. We call these methods *threshold-based*.

**Semantics Preserving Transformations** Data contamination not only occurs when including unmodified benchmark data in the training set, but also when including semantically equivalent samples. However, most contamination detection methods assume that benchmark data is included verbatim in the training data, or only allow for minimal perturbations such as extra newlines. While this is a fair assumption for the pretraining stage, even honest-but-negligent actors might use data augmentation techniques such as back-translation (Edunov et al., 2018) or paraphrasing (Li et al., 2018) during finetuning. To address this issue, Yang et al. (2023) propose to use an LLM to detect rephrased samples in the training data when given oracle access. Shi et al. (2023) propose a perplexity-based grey-box method which they observe to be robust under some paraphrasing, although it fails in our experiments (see Fig. 2).

## 5 EVADING CONTAMINATION DETECTION

We build on our analysis in §4 to characterize the requirements for successful evasion of data contamination detection. In particular, such a strategy should allow a malicious actor to significantly increase model performance on a benchmark while evading detection. Based on these requirements, we propose a simple technique to effectively evade contamination detection.

**Requirements** We identify the following requirements for a successful evasion strategy:

- Access: The strategy should be effective against all access levels, but can differ between assumed levels.

- Metadata: The strategy should remove all metadata to make the corresponding detection methods categorically ineffective.
- Reference Models: Despite reference models often being unavailable, a strong evasion strategy should be effective against reference- and threshold-based detection methods.
- Semantics Preserving Transformations: The strategy can alter the training data as long as it still enhances benchmark performance.

**Finetuning vs Pretraining**  Introducing data contamination during finetuning, rather than pretraining, is more advantageous for several reasons. First, optimizing only the conditional probability of the answer given the question prevents the model from memorizing the question, making detection much harder. Second, since the model sees less data after contamination, there is a lower chance of unlearning memorized samples, making performance gains more likely. Third, finetuning is significantly cheaper than pretraining, making it easier to implement and evaluate evasion strategies.

## 5.1 Evasion Through Rephrasing

As discussed in §4, most detection methods assume that the model is directly trained on the contaminated data, ignoring possible preprocessing contamination. This is a critical oversight that is trivial to exploit by rephrasing the benchmark data during preprocessing. We rephrase the benchmark data using GPT-4 (OpenAI, 2023) and finetune a pretrained model on a mix of background data unrelated to the benchmark and the rephrased benchmark data. Since we apply the strategy in the finetuning setting, metadata is naturally removed during the preprocessing stage. The rephrasing strategy varies based on the detection method's access level, with particular caution required for oracle access methods. We discuss and evaluate our evasion strategy for white-box access here, and defer the discussion of our evasion strategy for oracle access to App. C.

**White-Box Access**  As most black-, grey-, and white-box methods are based on the model reproducing benchmark data verbatim or assigning unusual perplexity, we expect them to be sensitive to semantics preserving rephrasing. Interestingly, rephrasing can also occur in the honest-but-negligent setting when rephrased samples are collected for the pre-training or finetuning data, or synthetically generated by a contaminated third-party language model. We use GPT-4 (OpenAI, 2023) to rephrase benchmark data using dataset-specific prompts provided in Fig. 3 of App. E.

## 6 Experiments

We demonstrate the effectiveness of rephrasing benchmark samples against contamination detection methods, showing it successfully reduces these methods to random guessing while still significantly increasing performance. We first describe our experimental setup (§6.1), then discuss the performance of contaminated models in various settings (§6.2), and finally demonstrate the effectiveness of our evasion strategy in evading current detection methods (§6.3 and §6.4). All reported confidence bounds are 2-sigma confidence intervals computed using bootstrapping over the samples in the benchmarks.

## 6.1 Experimental Setup

Below, we describe our general experimental setup, referring to App. E for more details.

**Benchmarks**  We select four popular benchmarks for evaluation: the math benchmark GSM8K (Cobbe et al., 2021), TruthfulQA (Lin et al., 2022) which contains questions on common misconceptions, and two multiple-choice question-answering datasets, ARC-Challenge (Clark et al., 2018) and a subset of MMLU (Hendrycks et al., 2021).

**Models**  We evaluate existing detection methods using Llama-3.1-8B (Dubey et al., 2024), Mistral-7B (Jiang et al., 2023), Phi-2 (Javaheripi et al., 2023), Phi-3-Small, and Phi-3.5-Mini (Abdin et al., 2024).

Table 1: Averaged accuracy in % of various models on various benchmarks under contaminated and uncontaminated settings. C (resp. U) is measured on the contaminated (resp. uncontaminated) part of the test set. 2-sigma intervals are shown in App. F and are about 2% for most values.

| | REFERENCE | | 1 OCCURRENCE | | | | 5 OCCURRENCES | | | |
| | | | OPEN | | EVASIVE | | OPEN | | EVASIVE | |
| | C | U | C | U | C | U | C | U | C | U |
|---|---|---|---|---|---|---|---|---|---|---|
| LLAMA-3.1-8B | 55.07 | 53.76 | 76.56 | 63.57 | 61.71 | 58.43 | 92.68 | 63.31 | 64.01 | 58.44 |
| MISTRAL-7B | 41.28 | 40.14 | 71.41 | 50.73 | 54.43 | 46.95 | 95.43 | 46.49 | 56.47 | 42.25 |
| PHI-2 | 43.00 | 41.45 | 65.39 | 49.69 | 52.71 | 47.38 | 85.61 | 52.20 | 58.12 | 47.60 |
| PHI-3-SMALL | 61.57 | 58.97 | 80.53 | 73.39 | 67.81 | 65.29 | 77.84 | 62.24 | 71.78 | 68.27 |
| PHI-3.5-MINI | 54.92 | 53.44 | 73.08 | 67.98 | 67.11 | 64.49 | 82.54 | 70.36 | 69.98 | 65.71 |

**Finetuning**  To compare results between the openly and evasively malicious settings, we finetune models on the instruction dataset OpenOrca (Lian et al., 2023) contaminated with a varying set of benchmark samples. Specifically, we include 50% of the original or rephrased benchmark data for the openly and evasively malicious settings, respectively, and repeat the contaminated portion of this data mixture one or five times during training. This leads to an effective contamination of 2% or 10% of the training set. In App. D, we analyze the performance of models trained on a benchmark with only 0.2% effective contamination, which is more typical for honest-but-negligent actors. We compare the performance to a model finetuned on the uncontaminated portion of our data.

## 6.2 PERFORMANCE OF CONTAMINATED MODELS

We report the performance of all finetuned models on the contaminated and uncontaminated half of the benchmark in Table 1. In the openly malicious setting, performance substantially increases across all benchmarks, improving the average accuracy on contaminated samples by 36% for five occurrences of the contaminated portion of the benchmark. Even on the uncontaminated samples, performance increases by 9% for five occurrences. This highlights that the common practice of honest-but-negligent actors to measure performance on a clean subset of the data (OpenAI, 2023; Touvron et al., 2023) can still lead to artificially inflated scores and is insufficient to obtain a representative performance estimate.

Performance improvement in the evasively malicious setting, while less pronounced, is still significant. Concretely, we observe an average gain of 10% and 13% on the contaminated samples for one and five occurrences, respectively, reduced to 7% on the uncontaminated samples. Thus, we conclude that while not as effective as finetuning on the original samples, finetuning on rephrased samples can significantly increase model performance both on the contaminated and uncontaminated portion of the test set and must therefore also be considered data contamination.

## 6.3 SAMPLE-LEVEL DETECTION METHODS

We evaluate several sample-level detection methods and present results in Table 2 for LLAMA-3.1-8B. In App. B we argue why it is sufficient to evaluate against these methods to claim evasion of all current detection methods by using our analysis in §4. While these methods show acceptable performance for openly malicious actors, they fail completely when using an evasive strategy for evasively malicious actors. We report similar results for all other models in App. F.

**Detection Methods**  We include seven sample-level detection methods that do not require oracle access or metadata contamination. Specifically, we consider the method proposed by Yeom et al. (2018), which measures the perplexity of a sample as a measure of memorization. We also include a method that instead measures the perplexity on the least likely k% of tokens (Shi et al., 2023) and various variants of this method (Zhang et al., 2024c; Zhang and Wu, 2024). Mireshghallah et al. (2022) computes a score for each sample by comparing the loss of the contaminated model to the loss of a reference model. Furthermore, we include a method by Carlini et al. (2021) that compares the loss of a sample to the loss of the same sample converted to lowercase text. Finally, we measure the performance of Xie et al. (2024), which measures contamination by comparing the perplexity of a sample with the perplexity of the same sample in a few-shot setting.

Table 2: Average TPR@5%FPR over the four benchmarks for various sample-level detection methods on LLAMA-3.1-8B. We compare openly malicious (OP) and evasively malicious (EV) actors.

| | 1 Occ. | | 5 Occ. | |
|---|---|---|---|---|
| | OP | EV | OP | EV |
| Mireshghallah et al. (2022) | $6.20_{\pm1.6}$ | $5.86_{\pm1.4}$ | $14.37_{\pm2.7}$ | $7.49_{\pm1.7}$ |
| Zhang and Wu (2024) | $6.59_{\pm1.5}$ | $4.29_{\pm1.4}$ | $20.44_{\pm2.6}$ | $5.09_{\pm1.5}$ |
| Zhang et al. (2024c) | $8.80_{\pm1.8}$ | $5.60_{\pm1.5}$ | $33.73_{\pm2.8}$ | $4.09_{\pm1.8}$ |
| Carlini et al. (2021) | $9.28_{\pm2.1}$ | $5.63_{\pm1.4}$ | $33.84_{\pm3.2}$ | $5.95_{\pm1.5}$ |
| Yeom et al. (2018) | $10.83_{\pm2.1}$ | $5.75_{\pm1.5}$ | $35.40_{\pm5.8}$ | $5.15_{\pm1.4}$ |
| Shi et al. (2023) | $11.65_{\pm2.1}$ | $5.50_{\pm1.6}$ | $36.86_{\pm5.4}$ | $5.05_{\pm1.3}$ |
| Xie et al. (2024) | $9.94_{\pm1.9}$ | $6.48_{\pm1.4}$ | $39.57_{\pm4.3}$ | $5.12_{\pm1.5}$ |

Table 3: Contamination score returned by Shi (2023) on the contaminated portion of the benchmarks for LLAMA-3.1-8B. We compare openly malicious (OP) and evasively malicious (EV) actors.

| | REF. | 1 Occ. | | 5 Occ. | |
|---|---|---|---|---|---|
| | | OP | EV | OP | EV |
| GSM8K | 0.14 | 0.67 | 0.11 | 0.82 | 0.12 |
| TruthfulQA | 0.30 | 0.27 | 0.15 | 0.42 | 0.16 |
| MMLU | 0.08 | 0.05 | 0.06 | 0.06 | 0.09 |
| ARC | 0.05 | 0.01 | 0.02 | 0.01 | 0.02 |

Table 4: CleanEval accuracy in % on the contaminated part of the benchmark. We compare openly malicious (OP) and evasively malicious (EV) actors. 2-sigma intervals are shown in App. F and are about 2% for most values.

| | REF. | 1 Occ. | | 5 Occ. | |
|---|---|---|---|---|---|
| | | OP | EV | OP | EV |
| LLAMA-3.1-8B | 54.8 | 73.1 | 61.1 | 82.1 | 63.5 |
| MISTRAL-7B | 42.3 | 68.8 | 53.1 | 82.5 | 57.2 |
| PHI-2 | 43.7 | 62.4 | 51.4 | 76.6 | 58.3 |
| PHI-3-SMALL | 60.4 | 76.5 | 68.6 | 68.8 | 69.6 |
| PHI-3.5-MINI | 56.0 | 68.2 | 63.8 | 74.5 | 66.8 |

**Results** We report the true positive rate (TPR) at a false positive rate of 5% (FPR) averaged over the four datasets in Table 2.. While the best prior method (Xie et al., 2024) achieves 10% and 40% TPR@5%FPR in the openly malicious setting depending on whether the contaminated data appears one or five times, respectively, no method achieves more than 7.5% TPR@5%FPR in the malicious setting even for five repeats of the contaminated data. As this barely beats random guessing, we conclude that all current sample-level detection methods fail to detect rephrased benchmark data. Interestingly, Zhang and Wu (2024); Zhang et al. (2024c) do not outperform Shi et al. (2023) despite being more complex methods that were based on this original method.

## 6.4 BENCHMARK-LEVEL DETECTION METHODS

We now evaluate benchmark-level detection methods, noting that most require metadata contamination (Golchin and Surdeanu, 2023a;b; Oren et al., 2023), which is absent in the finetuning setting. Therefore, we evaluate the only two methods that do not require metadata contamination: *DPCC* (Shi, 2023) and *CleanEval* (Zhu et al., 2023).

**DPCC** To detect contamination, Shi (2023) first generates several perturbations of a benchmark sample using a reference model and then measures the perplexity of the perturbed samples with the model under consideration. A contamination score is then computed based on the ratio of perturbed samples that have a higher perplexity than the unperturbed sample. This method is generally popular in the community and used on models related to the Open LLM Leaderboard (Beeching et al., 2023).

We report results for LLAMA-3.1-8B in Table 3 and observe that the contamination score is highly benchmark-dependent, reaching values between 0.05 for ARC and 0.30 for TruthfulQA in the absence of contamination. More importantly, the contamination threshold set by Shi (2023), 0.85, does not flag any model, making the method ineffective for benchmark-level contamination detection. However, we note a significant increase in contamination scores for GSM8k in the openly malicious setting, making detection possible for a lower threshold. In contrast, the scores in the evasively malicious setting on GSM8k are comparable to those of the uncontaminated model. The contamination scores for MMLU and ARC are small across all models, making detection unfeasible for these benchmarks. We obtain similar results for all other models and report them in App. F.

**CleanEval** CleanEval (Zhu et al., 2023) evaluates a model on a rephrased version of the benchmark to obtain an accurate comparison between contaminated and uncontaminated models. As the

exact rephrasing technique is not fully specified in their paper, we implement our variant which we describe in App. E. We report the performance on the rephrased benchmark in Table 4.

We find that contaminated models continue to outperform the uncontaminated baseline by a substantial margin, thus failing to provide accurate model comparisons. However, in the openly malicious setting, the performance gap is generally reduced by $5\%$ to $15\%$, indicating that CleanEval can detect the contamination. In the evasively malicious setting, the performance gap remains unchanged, thereby failing to detect any contamination.

## 7 RELATED WORK

The current practice among model providers to train language models in an honest-but-negligent fashion, combined with the risk of malicious actors actively contaminating models to achieve top benchmark performance, can make traditional benchmarks an unreliable indicator of model quality. We discuss several alternatives that circumvent the issues associated with static benchmarks, while still allowing for a comprehensive and reliable evaluation of the models.

**Dynamic Benchmarks** The static nature of traditional benchmarks is what allows both honest-but-negligent and malicious contamination to occur. Therefore, a recent line of work (Huang et al., 2023; Jain et al., 2024; Li and Flanigan, 2023; Li et al., 2023; Roberts et al., 2023; Shi et al., 2023; Zhang et al., 2024b) has focused on evaluation using *dynamic benchmarks*. Specifically, as dynamic benchmarks are periodically updated, they allow measuring model performance on data that was not available during training. Further, these benchmarks can compare performance on data collected before and after a model release and thus provide a simple way to measure contamination.

However, their dynamic nature comes with significant challenges: First, high-quality benchmarks take considerable time and effort to create, leading to most dynamic benchmarks being less well-curated than traditional benchmarks. Second, performance on dynamic benchmarks can vary over time, making it harder to track progress and compare models. Especially the possibility of new models training on prior versions of the benchmark can lead to a false sense of progress. Third, continued effort is required to ensure that the benchmark is continuously updated.

**Human Evaluation** Human evaluations allow for comprehensive model evaluation with limited risk of contamination over a wide range of tasks (Chang et al., 2023; Freitag et al., 2021; Zheng et al., 2023). However, it is both time-consuming and expensive, requiring a large number of expert evaluators and a good experimental setup to prevent human biases from influencing results (Chang et al., 2023; Zheng et al., 2023). Further, human preferences can differ between individuals, and groups (Peng et al., 1997). While crowd-sourcing initiatives like Zheng et al. (2023) can help mitigate these issues, they are vulnerable to attacks by malicious actors aiming to boost their ratings.

**Private Benchmarks** Benchmark contamination can be avoided by preventing model providers from accessing the benchmark data. These *private benchmarks* (Zhang et al., 2024a) would prevent model providers from accidentally or maliciously including the benchmark in their training data and therefore provide the possibility for reliable model evaluation. This approach would need careful consideration and continuous monitoring since any data leakage would effectively negate the benefits. Furthermore, evaluation cannot be performed by the model provider, as this would inevitably leak the benchmark data. This poses a significant challenge for closed-source models, which do not share any model specifics (Anil et al., 2023a; OpenAI, 2023) and therefore need provable guarantees that these specifics do not get leaked.

## 8 LIMITATIONS

While simple and effective, we note that there are several limitations to contamination through rephrasing. First, it does not yield the same performance improvement as training on the actual benchmark samples, indicating that there is still potential to improve the attack. Second, the attack is limited to public benchmarks. As discussed in §7, there are other types of benchmarks that we cannot attack due to the lack of available public data for training purposes. This limitation raises an important question about the security of these paradigms against malicious model providers. We believe this question offers a promising direction for future research.

## 9 CONCLUSION

In this work, we discussed the importance of considering malicious model providers that actively contaminate their models to achieve artificially high performance on specific benchmarks. Our analysis of contamination detection methods, focusing on their fundamental assumptions, revealed critical shortcomings when confronted with evasively malicious actors. We exploit these shortcomings with a simple contamination technique that evades detection by rephrasing samples while increasing performance on public benchmarks by up to 15%.

## 10 ETHICS STATEMENT

Public benchmarks are essential for evaluating the performance of language models. Our work demonstrates the potential for malicious actors to actively contaminate the training data while evading detection, highlighting a significant security concern. By discussing the possibility of malicious actors, we aim to raise awareness about the issue and encourage the development of more robust evaluation methods. More importantly, since the vulnerability discussed in this work is very simple and the stakes in this competitive field are very high, we believe it is possible that malicious actors are already exploiting similar vulnerabilities. However, there is a risk that our findings are exploited, further compromising the reliability of public benchmarks. Despite these concerns, we believe that publishing our results is beneficial, as the worst-case scenario is the adoption of suboptimal models for specific tasks.

## 11 REPRODUCIBILITY STATEMENT

We provide the code to reproduce all our experiments in the supplementary material. This code is clearly documented and contains all the necessary instructions to run the experiments and reproduce our results.

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

## A    ATTRIBUTION

We provide attribution for the icons used in Fig. 1. The golden, silver and bronze medals are by Md Tanvirul Haque. The red flag is by Alfredo Hernandez.

## B    ASSUMPTION FOR DETECTION METHODS

We present a table with an overview of the discussed assumptions on data contamination detection from §4 in Table 5. We no argue why the inclusion of the methods in our experiments is sufficient to claim evasion of all contamination detection methods.

Based on our analysis presented in §4, we can exclude the following methods from this table for our evaluation:

- **Methods requiring oracle access:** Techniques that require oracle-level access are addressed separately in App. C, where we demonstrate that our evasion strategy works effectively even with oracle-level attacks. For the attack evaluated in the main part of the paper, we can exclude them.

- **Metadata-dependent detection methods:** Methods relying on metadata are irrelevant because we explicitly remove metadata from contaminated samples. This ensures that detection mechanisms dependent on metadata are not applicable to our approach.

- **Methods analyzing contamination with no novel method:** Li (2023b); Xu et al. (2024); Carlini et al. (2022) primarily offer analyses or conceptual discussions rather than detection strategies. As such, they do not introduce a new method that needs to be evaluated in our experiments.

- **Duplicate techniques:** Some works duplicate existing detection methods already included in our evaluation. For example, Watson et al. (2022) is a reimplementation of (Mireshghallah et al., 2022), and (Li, 2023a) is functionally equivalent to (Yeom et al., 2018).

- **Wrong-Option Contamination:** Deng et al. (2023) requires the inclusion of incorrect options of a benchmark in the training set, which we explicitly remove during the rephrasing process. In some sense, wrong options are a form of metadata.

- **User-Level Contamination Methods:** Song and Shmatikov (2019) address user-level contamination in a privacy-based use-case. Therefore, they do not apply to the contamination scenario studied in this work.

After excluding the above methods, we remain with the methods that we evaluate in our experiments. Our results in §6 demonstrate that our attack effectively circumvents these methods by reducing their performance to random chance.

Table 5: Overview of prior work on data contamination. In the level column, S stands for sample-level and B for benchmark-level. In the access column, O stands for oracle access, B for black-box access and G for grey-box access. $\checkmark^*$ in the reference columns indicates the method is threshold-based instead of reference-based.

| Method | Level | Access | Metadata | Reference | Verbatim |
|---|---|---|---|---|---|
| Dodge et al. (2021) | S | O | ✗ | ✗ | ✓ |
| Brown et al. (2020) | S | O | ✗ | ✗ | ✓ |
| Chowdhery et al. (2022) | S | O | ✗ | ✗ | ✓ |
| Touvron et al. (2023) | S | O | ✗ | ✗ | ✓ |
| OpenAI (2023) | S | O | ✗ | ✗ | ✓ |
| Vu et al. (2023) | S | O | ✗ | ✗ | ✓ |
| Yang et al. (2023) | S | O | ✗ | ✗ | ✗ |
| Mattern et al. (2023) | S | G | ✗ | ✗ | ✓ |
| Li (2023b) | S | B | ✗ | ✗ | ✓ |
| Deng et al. (2023) | S | B | ✗ | ✗ | ✓ |
| Zhu et al. (2023) | B | B | ✗ | ✗ | ✓ |
| Xu et al. (2024) | B | B & G | ✗ | ✗ | ✓ |
| Oren et al. (2023) | B | G | ✓ | ✗ | ✓ |
| Golchin and Surdeanu (2023a) | B | B | ✓ | ✗ | ✓ |
| Golchin and Surdeanu (2023b) | S & B | B | ✓ | ✗ | ✓ |
| Duarte et al. (2024) | S | B | ✓ | ✗ | ✓ |
| Song and Shmatikov (2019) | S | B | ✗ | ✓ | ✓ |
| Watson et al. (2022) | S | G | ✗ | ✓ | ✓ |
| Carlini et al. (2022) | S | G | ✗ | ✓ | ✓ |
| Mireshghallah et al. (2022) | S | G | ✗ | ✓ | ✓ |
| Shi (2023) | B | G | ✗ | ✓ | ✓ |
| Xie et al. (2024) | B | G | ✗ | ✓ | ✓ |
| Zhang and Wu (2024) | B | G | ✗ | ✓ | ✓ |
| Zhang et al. (2024c) | B | G | ✗ | ✓ | ✓ |
| Li (2023a) | B | G | ✗ | $\checkmark^*$ | ✓ |
| Carlini et al. (2021) | S | G | ✗ | $\checkmark^*$ | ✓ |
| Yeom et al. (2018) | S | G | ✗ | $\checkmark^*$ | ✓ |
| Shi et al. (2023) | S | G | ✗ | $\checkmark^*$ | ✗ |

## C  EVASION FOR ORACLE ACCESS

We discuss how we can adjust our evasion strategy to evade oracle access detection and evaluate its
.

**Evasion for Oracle Access**  The default rephrasing we use is frequently unable to evade oracle access methods. For example, Yang et al. (2023) explicitly ask a language model if a sample from the training data is a rephrased version of a benchmark sample. While this technique requires effective prefiltering to become computationally feasible for large datasets, we can still evade it by more aggressively rephrasing the benchmark samples. To this end, we iteratively rephrase a sample and request GPT-4 to verify if it has been unrecognizably rephrased, guiding it towards more significant rephrasing each time. We find that even a few iterations of this are highly effective at evading oracle access methods while still significantly increasing performance. Further, we note that we can drop all samples that are still detected from the training data.

### C.1  ORACLE ACCESS DETECTION METHODS

We evaluate our method against two oracle access detection methods. First, we consider an n-gram overlap check (Brown et al., 2020; Touvron et al., 2023), using the most aggressive criterion for

Table 6: Detection rate in % of oracle access detection methods using advanced rephrasing.

|  | Yang et al. (2023) | N-gram |
|---|---|---|
| GSM8K | 21.4 | 0.7 |
| TruthfulQA | 50.2 | 0.1 |
| MMLU | 11.9 | 0.7 |
| ARC | 28.9 | 0.1 |

Table 7: Accuracy in % on various benchmarks using oracle rephrasing. C is measured on the contaminated part of the test set, U on the uncontaminated part of the test set.

|  | REFERENCE | | 1 OCC. | | 5 OCC. | |
|---|---|---|---|---|---|---|
|  | C | U | C | U | C | U |
| LLAMA-3.1-8B | $55.44_{\pm2.1}$ | $53.45_{\pm2.0}$ | $59.45_{\pm2.1}$ | $58.78_{\pm2.0}$ | $58.87_{\pm2.1}$ | $58.06_{\pm2.1}$ |
| MISTRAL-7B | $41.09_{\pm2.0}$ | $40.30_{\pm2.0}$ | $47.86_{\pm2.1}$ | $44.58_{\pm2.1}$ | $44.34_{\pm2.1}$ | $43.23_{\pm2.1}$ |
| PHI-2 | $43.94_{\pm2.1}$ | $40.58_{\pm2.0}$ | $48.29_{\pm2.1}$ | $45.86_{\pm2.1}$ | $51.91_{\pm2.1}$ | $46.48_{\pm2.1}$ |
| PHI-3-SMALL | $59.97_{\pm2.1}$ | $60.59_{\pm2.0}$ | $63.88_{\pm2.0}$ | $63.41_{\pm2.0}$ | $68.09_{\pm2.0}$ | $66.28_{\pm2.0}$ |
| PHI-3.5-MINI | $54.54_{\pm2.1}$ | $53.82_{\pm2.1}$ | $65.30_{\pm2.0}$ | $65.45_{\pm2.0}$ | $67.19_{\pm2.0}$ | $65.81_{\pm2.0}$ |

contamination we are aware of, a single 8-gram overlap (Touvron et al., 2023). Second, we evaluate the stronger oracle access detection method proposed by Yang et al. (2023), *LLM Decontaminator*, which leverages an LLM to check if two samples are rephrased versions of each other. We note that our method for white-box access can be detected by LLM Decontaminator in over 97% of cases.

Specifically, we ask GPT-4 to make further significant changes to the already rephrased sample and then only include samples in the training set that successfully evade detection. In Table 6 we report the detection rates of the aggressively rewritten samples prior to this filtering.

**Detection Rate**   As expected, we find that the traditional n-gram method is generally ineffective, flagging less than 1% of the contaminated data. LLM Decontaminator is much more effective, detecting up to half of the rephrased samples. However, by dropping all flagged samples from our training set, we can still perfectly evade even this oracle access method. We note that a second round of strong rephrasing on the TruthfulQA dataset further reduces the detection rate from 50% to 25%, showing that consecutive rephrasing can be employed to use a greater amount of samples during training if necessary.

**Performance**   We report the performance of models finetuned on a data mixture consisting of OpenOrca combined with the heavily rephrased data that was not flagged by either oracle access method in Table 7. We find that training on the rephrased benchmark still significantly improves performance and can therefore be considered problematic. Specifically, contaminated samples show an accuracy increase up to 11% and 13% for one and five occurrences, respectively. Thus, we conclude that current oracle access detection methods are insufficient to detect rephrased samples.

## D   CONTAMINATION FOR HONEST-BUT-NEGLIGENT ACTORS

Honest-but-negligent actors are more likely to introduce a smaller amount of contamination compared to the setting presented in §6.2. We therefore perform an additional experiment where only 5% of the benchmark data is contaminated, and this benchmark data only occurs once. This corresponds to only 0.2% of the training data being contaminated. We use the same benchmarks as in §6.2 and report the results in Table 8.

While performance on the contaminated set has a much higher variance due to its smaller size, we still observe clear increases in performance for both the MMLU and ARC benchmarks. Performance on the uncontaminated samples is increased notably less with only ARC showing significant increases in the openly contaminated setting. Interestingly, the evasively malicious setting seems to have improved in-benchmark generalization leading to significant improvements on the uncontaminated portion of GSM8K, MMLU, and ARC. This suggests that the evasively malicious setting

Table 8: Performance of Phi-2 on various benchmarks under contaminated and uncontaminated settings. The metric used is accuracy in %. C is measured on the (small) contaminated part of the test set, U on the (large) uncontaminated part of the test set.

| | REFERENCE | | OPEN | | EVASIVE | |
|---|---|---|---|---|---|---|
| | C | U | C | U | C | U |
| GSM8K | $21.2_{\pm 10.0}$ | $24.9_{\pm 2.3}$ | $19.7_{\pm 9.7}$ | $25.3_{\pm 2.4}$ | $34.8_{\pm 11.2}$ | $37.3_{\pm 2.7}$ |
| TruthfulQA | $43.9_{\pm 14.6}$ | $42.9_{\pm 3.4}$ | $48.8_{\pm 15.5}$ | $41.0_{\pm 3.5}$ | $39.0_{\pm 14.7}$ | $43.3_{\pm 3.4}$ |
| MMLU | $44.0_{\pm 14.1}$ | $43.6_{\pm 3.2}$ | $76.0_{\pm 12.4}$ | $43.6_{\pm 3.1}$ | $50.0_{\pm 13.8}$ | $48.3_{\pm 3.3}$ |
| ARC | $50.8_{\pm 12.5}$ | $58.0_{\pm 3.0}$ | $74.6_{\pm 10.8}$ | $63.7_{\pm 2.9}$ | $66.1_{\pm 11.8}$ | $65.4_{\pm 2.8}$ |

might be more sample-efficient for training models, as the model is forced to learn more generalizable features to perform well on the contaminated samples.

# E  EXPERIMENTAL DETAILS

We describe the experimental details of our experiments performed in §6. Specifically, we discuss the prompts, finetuning parameters, and preprocessing steps used for each step. Each finetuned model took around two hours on a single Nvidia H100 GPU. All experiments were performed using around six weeks of computation on a single Nvidia H100 GPU.

**Benchmarks**   For each benchmark, we only select the test data for evaluation. Furthermore, for the MMLU benchmark, we only select samples from the alphabetically first seven domains, to ensure that the benchmark is similar in size as the other benchmarks. Specifically, we select the `abstract algebra`, `anatomy`, `astronomy`, `business ethics`, `clinical knowledge`, `college biology` and `college chemistry` domains.

**Rephrasing**   We use GPT-4 (OpenAI, 2023) with a temperature of $0$ as the model with which we rephrase. This allows us to generate human-level quality rephrases. For each benchmark, we use a slightly adapted system prompt to generate rephrases. All system prompts are presented in Fig. 3. The user input is formatted as follows:

```
User Prompt

   Question: {{question}}
   Answer: {{answer}}
```

In order to avoid the detection method that requires memorization of wrong options (Deng et al., 2023), wrong options are omitted for MMLU and ARC-Challenge.

For oracle rephrasing, we continue from the rephrased question and answer and tell the model its rephrasing should be further adjusted. The prompts used to do so for each benchmark are presented in Fig. 4.

**Data Preparation**   For each benchmark, we randomly select 50% of the samples that are used when training on benchmark data. Depending on the setting, we then either copy the (rephrased) benchmark data one or five times and pad the resulting training data with randomly selected samples from the OpenOrca instruction-tuning dataset (Lian et al., 2023) until there are 25000 samples in the dataset. We note that the randomly selected samples from OpenOrca are mostly the same for all settings, with the minor difference that fewer samples are selected when the benchmark data is copied five times. We format prompts using the Alpaca formatting convention (Taori et al., 2023). Specifically, we use the following format for each sample:

**Prompt Format**

```
### Instruction:
{{instruction}}

### Input:
{{input}}

### Response:
{{response}}
```

If no instruction is available (which is the case for all benchmark data), we omit the instruction.

**Finetuning**   We use the HuggingFace Transformers library (Wolf et al., 2019) to finetune models. Specifically, we do full finetuning of each model with the default optimizer and a learning rate of $7 \cdot 10^{-5}$ for PHI-2 and $10^{-5}$ for all other models. Additionally, we use a warmup ratio of $0.05$ and use a batch size of $16$ in all settings. We finetune each model on a single epoch of the training data (possibly including up to 5 copies of the benchmark data).

**Performance Evaluation**   We evaluate the accuracy of each model on the test set of each benchmark in the zero-shot setting. For GSM8K, we parse the final number in the generated answer and compare it to the one in the output. For TruthfulQA, we compare the lowest perplexity of the model on the correct answers compared to the lowest perplexity on the incorrect answers and count a question as correct if the former is lower than the latter. For both MMLU and ARC, we first allow the model to generate an answer. We then select the option that has the highest ROUGE-L overlap (Lin, 2004) when compared with the generated answer.

**Detection**   For most detection methods, we use either the code for the method or our own implementation of the described method as described in the respective papers using the default parameters. For Shi (2023) we use the base model as the reference model.

Only regarding CleanEval (Zhu et al., 2023), do we diverge a bit from the method described in the paper. The authors described a rephrasing method that consists of three phases. First, they paraphrase samples using either language models or back-translation. Then, they filter the resulting data to ensure semantic equivalence between the original and rephrased sample. Finally, they select a sample that has the lowest BLEURT overlap score (Sellam et al., 2020) with the original sample. Unfortunately, it is not clear (1) how back-translation was done (which languages, how often, and which model), (2) what prompt was used for paraphrasing and (3) how many candidate samples were generated before filtering. Since we believe GPT-4 can accurately rephrase samples, and since the authors show that solely paraphrasing results in a dataset with comparable BLEURT-score as their dataset (Zhu et al. (2023), table 3), we only perform paraphrasing with GPT-4 and use the following system prompt for GSM8K and TruthfulQA:

**System Prompt CleanEval GSM8K and TruthfulQA**

```
Significantly rephrase the given question, but make sure the answer is
still the same. Do not include the answer in your response.

Format your reply as:
New Question: [New rephrased question]
```

and use the following system prompt for ARC-Challenge and MMLU:

**System Prompt CleanEval MMLU and ARC**

```
Significantly rephrase the given question and options, but make sure that
all possible options still have the same label. Label the multiple choice
```

Table 9: Averaged accuracy in % of various models on various benchmarks under contaminated and uncontaminated settings. C (resp. U) is measured on the contaminated (resp. uncontaminated) part of the test set. 2-sigma intervals are shown.

| | REFERENCE | | 1 OCCURRENCE | | | | 5 OCCURRENCES | | | |
| | | | OPEN | | EVASIVE | | OPEN | | EVASIVE | |
| | C | U | C | U | C | U | C | U | C | U |
|---|---|---|---|---|---|---|---|---|---|---|
| LLAMA-3.1-8B | $55.07_{\pm2.1}$ | $53.76_{\pm2.1}$ | $76.56_{\pm1.7}$ | $63.57_{\pm2.0}$ | $61.71_{\pm2.0}$ | $58.43_{\pm2.1}$ | $92.68_{\pm1.0}$ | $63.31_{\pm2.0}$ | $64.01_{\pm2.0}$ | $58.44_{\pm2.1}$ |
| MISTRAL-7B | $41.28_{\pm2.0}$ | $40.14_{\pm2.0}$ | $71.41_{\pm1.7}$ | $50.73_{\pm2.0}$ | $54.43_{\pm2.0}$ | $46.95_{\pm2.1}$ | $95.43_{\pm0.9}$ | $46.49_{\pm2.1}$ | $56.47_{\pm2.1}$ | $42.25_{\pm2.0}$ |
| PHI-2 | $43.00_{\pm2.1}$ | $41.45_{\pm2.1}$ | $65.39_{\pm2.0}$ | $49.69_{\pm2.1}$ | $52.71_{\pm2.1}$ | $47.38_{\pm2.1}$ | $85.61_{\pm1.3}$ | $52.20_{\pm2.1}$ | $58.12_{\pm2.1}$ | $47.60_{\pm2.1}$ |
| PHI-3-SMALL | $61.57_{\pm2.1}$ | $58.97_{\pm2.1}$ | $80.53_{\pm1.7}$ | $73.39_{\pm1.9}$ | $67.81_{\pm2.0}$ | $65.29_{\pm2.0}$ | $77.84_{\pm1.6}$ | $62.24_{\pm1.9}$ | $71.78_{\pm1.9}$ | $68.27_{\pm2.0}$ |
| PHI-3.5-MINI | $54.92_{\pm2.1}$ | $53.43_{\pm2.2}$ | $73.08_{\pm1.8}$ | $67.98_{\pm2.0}$ | $67.11_{\pm2.0}$ | $64.48_{\pm2.1}$ | $82.54_{\pm1.6}$ | $70.36_{\pm1.9}$ | $69.98_{\pm1.9}$ | $65.71_{\pm2.0}$ |

```
answers with A:, B:, C:, D:, E:. Do not include the answer in your response
.

Format your reply as:
New Question: [New rephrased question]
```

We format the user prompt as:

**User Prompt**

```
Question: {{question}}
Answer: {{answer}}
```

and include the options in the question.

We note that this is a different prompt from the one used for rephrasing in our experiments. Since the rephrased setting does not have a significant increase in performance compared to the uncontaminated baseline, as shown in Table 4, we assume that the potential correlation between two different rephrases of GPT-4 has no effect on our results.

## F   DETAILED RESULTS

We report the 2-sigma intervals for all experiments in §6. Furthermore, we report all results related to models that were not shown in §6.

The equivalent of Table 1 is shown in Table 9 and the equivalent of Table 4 is shown in Table 10, both now containing the complete confidence intervals. Tables 11–14 show the results of various sample-level detection methods on MISTRAL-7B, PHI-2, PHI-3-SMALL, and PHI-3.5-MINI, respectively. Tables 15–18 show the results of the benchmark-level detection method by Shi (2023) on MISTRAL-7B, PHI-2, PHI-3-SMALL, and PHI-3.5-MINI, respectively.

In Tables 19–23, we report detailed results for all models on all benchmarks. We find that the evasively malicious actor consistently outperforms the baseline model across all models and benchmarks. However, performance gains tend to vary between models and benchmarks: while LLAMA-3.1-8B, MISTRAL-7B, and PHI-2 show their most significant gains for GSM8k and TruthfulQA, PHI-3-SMALL and PHI-3.5-MINI show their most significant gains for MMLU and ARC-Challenge. Furthermore, there is one notable exception to the consistent gain across all benchmarks: the evasively malicious models of PHI-3.5-MINI and PHI-3-SMALL underperforms the baseline model for the TruthfulQA dataset. As the openly malicious model also underperforms the baseline model in the case of PHI-3.5-MINI, this suggests that the finetuning process is less effective for these models and dataset.

## G   LICENSING INFORMATION

We include the license for all models, benchmarks and other assets used in this paper in Table 24.

**System Prompt GSM8K**

```
You are a helpful assistant. The user will give you a question and answer
from the gsm8k dataset. Rewrite the question and answer. Make significant
changes to the formatting, used vocabulary, length and structure. Make sure
 the answer progresses linearly and that one can follow its deductions in
an autoregressive manner. Ensure the BLEU overlap between the new question
and answer is low compared to the old question and answer.

Format your reply as:
Reasoning: [brief reasoning on how to best rewrite and restructure question
 and answer]
New Question: [New rephrased question]
New Answer: [New rephrased answer]
```

**System Prompt TruthfulQA**

```
You are a helpful assistant. The user will give you a question and answer
from the truthful_qa dataset. Rephrase both the question and answer. Make
significant changes to used vocabulary, length and structure.

Format your reply as:
Reasoning: [brief reasoning on how to best rewrite and restructure question
 and answer]
New Question: [New rephrased question]
New Answer: [New rephrased answer]
```

**System Prompt MMLU**

```
You are a helpful assistant. The user will give you a question and answer
from the MMLU dataset. Rewrite both the question and answer. Make
significant changes to used vocabulary, length and structure. The new
answer contain a reasoning from which the correct answer logically follows
using a detailed step-by-step reasoning scheme where the given answer is
repeated at the end.

Format your reply as:
Reasoning: [brief reasoning on how to best rewrite and restructure question
 and answer]
New Question: [New rephrased question]
New Answer: [New rephrased answer]
```

**System Prompt ARC**

```
You are a helpful assistant. The user will give you a question and answer
from the ARC-Challenge dataset. Rephrase both the question and answer. Make
 significant changes to used vocabulary, length and structure.

Format your reply as:
Reasoning: [brief reasoning on how to best rewrite and restructure question
 and answer]
New Question: [New rephrased question]
New Answer: [New rephrased answer]
```

Figure 3: System prompts used for rephrasing.

**User Prompt GSM8K**

```
Rewrite the question and answer further such that the background story,
names and numbers are completely different. Make sure it is difficult to
recognize that one is a rewrite of the other. Use the same reply format.
```

**User Prompt TruthfulQA**

```
A human could still detect that the new question and answer are based on
the original ones. Make significant changes to the question and change the
discussed misconception in order to make such an observation impossible.
Use the same format.
```

**User Prompt MMLU**

```
A human could still detect that the new question and answer are based on
the original ones. Make very significant changes to the question and answer
 to make such an observation completely impossible. Change numbers,
background story and all you can change to make this happen. Use the same
format.
```

**User Prompt ARC**

```
A human could still detect that the new question and answer are based on
the original ones. Make very significant changes to the question and answer
 to make such an observation completely impossible. Change numbers,
background story and all you can change to make this happen. Use the same
format.
```

Figure 4: User prompts used for further rephrasing of each benchmark.

Table 10: CleanEval accuracy in % on the contaminated part of the benchmark. We compare openly malicious (OP) and evasively malicious (EV) actors. 2-sigma intervals are shown.

| | REF. | 1 OCC. | | 5 OCC. | |
|---|---|---|---|---|---|
| | | OP | EV | OP | EV |
| LLAMA-3.1-8B | $54.84_{\pm 2.1}$ | $73.11_{\pm 1.8}$ | $61.09_{\pm 2.1}$ | $82.10_{\pm 1.5}$ | $63.48_{\pm 2.0}$ |
| MISTRAL-7B | $42.34_{\pm 2.0}$ | $68.80_{\pm 1.8}$ | $53.05_{\pm 2.0}$ | $82.53_{\pm 1.5}$ | $57.15_{\pm 2.1}$ |
| PHI-2 | $43.73_{\pm 2.1}$ | $62.36_{\pm 2.0}$ | $51.42_{\pm 2.1}$ | $76.59_{\pm 1.7}$ | $58.31_{\pm 2.1}$ |
| PHI-3-SMALL | $60.43_{\pm 2.0}$ | $76.55_{\pm 1.9}$ | $68.59_{\pm 2.0}$ | $68.85_{\pm 1.6}$ | $69.61_{\pm 2.0}$ |
| PHI-3.5-MINI | $55.96_{\pm 2.1}$ | $68.16_{\pm 2.0}$ | $63.82_{\pm 2.0}$ | $74.49_{\pm 1.8}$ | $66.76_{\pm 2.0}$ |

Table 11: Average TPR@5%FPR over the four benchmarks for various sample-level detection methods on MISTRAL-7B. We compare openly malicious (OP) and evasively malicious (EV) actors.

| | 1 OCC. | | 5 OCC. | |
|---|---|---|---|---|
| | OP | EV | OP | EV |
| Mireshghallah et al. (2022) | $6.76_{\pm 1.6}$ | $6.53_{\pm 1.5}$ | $21.27_{\pm 3.5}$ | $7.50_{\pm 1.4}$ |
| (Zhang and Wu, 2024) | $5.83_{\pm 1.4}$ | $4.47_{\pm 1.2}$ | $20.20_{\pm 2.9}$ | $5.97_{\pm 1.5}$ |
| (Zhang et al., 2024c) | $11.32_{\pm 2.4}$ | $5.29_{\pm 1.3}$ | $18.82_{\pm 1.4}$ | $4.34_{\pm 1.2}$ |
| Carlini et al. (2021) | $11.16_{\pm 2.0}$ | $4.69_{\pm 1.3}$ | $29.70_{\pm 2.3}$ | $4.77_{\pm 1.4}$ |
| Yeom et al. (2018) | $12.54_{\pm 2.5}$ | $5.17_{\pm 1.3}$ | $33.92_{\pm 3.7}$ | $5.21_{\pm 1.3}$ |
| Shi et al. (2023) | $13.42_{\pm 2.4}$ | $4.99_{\pm 1.3}$ | $32.68_{\pm 3.1}$ | $4.97_{\pm 1.4}$ |
| (Xie et al., 2024) | $4.72_{\pm 1.7}$ | $5.93_{\pm 1.4}$ | $14.76_{\pm 2.0}$ | $4.66_{\pm 1.2}$ |

Table 12: Average TPR@5%FPR over the four benchmarks for various sample-level detection methods on PHI-2. We compare openly malicious (OP) and evasively malicious (EV) actors.

| | 1 OCC. | | 5 OCC. | |
|---|---|---|---|---|
| | OP | EV | OP | EV |
| Mireshghallah et al. (2022) | $6.92_{\pm 1.6}$ | $5.68_{\pm 1.4}$ | $12.40_{\pm 2.2}$ | $7.20_{\pm 1.5}$ |
| (Zhang and Wu, 2024) | $5.13_{\pm 1.4}$ | $4.54_{\pm 1.3}$ | $5.25_{\pm 1.5}$ | $4.89_{\pm 1.4}$ |
| (Zhang et al., 2024c) | $7.98_{\pm 1.7}$ | $4.41_{\pm 1.4}$ | $25.45_{\pm 3.2}$ | $5.04_{\pm 1.4}$ |
| Carlini et al. (2021) | $10.98_{\pm 2.1}$ | $5.01_{\pm 1.5}$ | $29.61_{\pm 3.2}$ | $4.17_{\pm 1.4}$ |
| Yeom et al. (2018) | $15.49_{\pm 2.4}$ | $5.25_{\pm 1.6}$ | $45.56_{\pm 4.9}$ | $5.40_{\pm 1.4}$ |
| Shi et al. (2023) | $15.18_{\pm 2.5}$ | $4.79_{\pm 1.4}$ | $46.11_{\pm 5.2}$ | $5.24_{\pm 1.4}$ |
| (Xie et al., 2024) | $9.31_{\pm 2.2}$ | $5.68_{\pm 1.3}$ | $21.71_{\pm 3.7}$ | $4.41_{\pm 1.3}$ |

Table 13: Average TPR@5%FPR over the four benchmarks for various sample-level detection methods on PHI-3-SMALL. We compare openly malicious (OP) and evasively malicious (EV) actors.

| | 1 OCC. | | 5 OCC. | |
|---|---|---|---|---|
| | OP | EV | OP | EV |
| Mireshghallah et al. (2022) | $6.49_{\pm 1.6}$ | $4.94_{\pm 1.6}$ | $10.07_{\pm 2.1}$ | $5.93_{\pm 1.5}$ |
| (Zhang and Wu, 2024) | $4.88_{\pm 1.3}$ | $5.30_{\pm 1.4}$ | $5.19_{\pm 1.2}$ | $4.52_{\pm 1.3}$ |
| (Zhang et al., 2024c) | $5.93_{\pm 1.7}$ | $4.79_{\pm 1.3}$ | $10.54_{\pm 2.7}$ | $4.69_{\pm 1.3}$ |
| Carlini et al. (2021) | $5.58_{\pm 1.4}$ | $3.82_{\pm 1.7}$ | $10.07_{\pm 2.4}$ | $4.37_{\pm 1.4}$ |
| Yeom et al. (2018) | $6.38_{\pm 1.6}$ | $4.38_{\pm 1.3}$ | $13.34_{\pm 2.4}$ | $4.71_{\pm 1.3}$ |
| Shi et al. (2023) | $6.97_{\pm 1.7}$ | $4.45_{\pm 1.3}$ | $13.83_{\pm 2.3}$ | $4.62_{\pm 1.3}$ |
| (Xie et al., 2024) | $5.52_{\pm 1.5}$ | $5.16_{\pm 1.3}$ | $11.74_{\pm 2.8}$ | $6.27_{\pm 1.4}$ |

Table 14: Average TPR@5%FPR over the four benchmarks for various sample-level detection methods on PHI-3.5-MINI. We compare openly malicious (OP) and evasively malicious (EV) actors.

|  | 1 Occ. | | 5 Occ. | |
| --- | --- | --- | --- | --- |
|  | OP | EV | OP | EV |
| Mireshghallah et al. (2022) | $6.12_{\pm1.4}$ | $5.72_{\pm1.4}$ | $6.75_{\pm1.4}$ | $6.08_{\pm1.5}$ |
| (Zhang and Wu, 2024) | $5.54_{\pm1.3}$ | $5.78_{\pm1.9}$ | $6.20_{\pm1.8}$ | $5.63_{\pm1.5}$ |
| (Zhang et al., 2024c) | $4.06_{\pm1.3}$ | $5.29_{\pm1.4}$ | $5.76_{\pm1.5}$ | $5.33_{\pm1.4}$ |
| Carlini et al. (2021) | $4.94_{\pm1.3}$ | $4.54_{\pm1.3}$ | $6.52_{\pm1.7}$ | $4.74_{\pm1.4}$ |
| Yeom et al. (2018) | $5.61_{\pm1.3}$ | $4.59_{\pm1.5}$ | $7.47_{\pm1.6}$ | $4.60_{\pm1.7}$ |
| Shi et al. (2023) | $6.01_{\pm1.5}$ | $4.57_{\pm1.4}$ | $6.94_{\pm1.6}$ | $4.42_{\pm1.5}$ |
| (Xie et al., 2024) | $4.77_{\pm1.5}$ | $4.84_{\pm1.2}$ | $6.34_{\pm1.8}$ | $6.04_{\pm1.2}$ |

Table 15: Contamination score returned by Shi (2023) on the contaminated portion of the benchmarks for MISTRAL-7B. We compare openly malicious (OP) and evasively malicious (EV) actors.

|  | REF. | 1 Occ. | | 5 Occ. | |
| --- | --- | --- | --- | --- | --- |
|  |  | OP | EV | OP | EV |
| GSM8K | 0.89 | 1.00 | 0.91 | 1.00 | 0.91 |
| TruthfulQA | 0.61 | 0.81 | 0.66 | 0.86 | 0.58 |
| MMLU | 0.22 | 0.21 | 0.33 | 0.20 | 0.42 |
| ARC | 0.10 | 0.10 | 0.14 | 0.11 | 0.16 |

Table 16: Contamination score returned by Shi (2023) on the contaminated portion of the benchmarks for PHI-2. We compare openly malicious (OP) and evasively malicious (EV) actors.

|  | REF. | 1 Occ. | | 5 Occ. | |
| --- | --- | --- | --- | --- | --- |
|  |  | OP | EV | OP | EV |
| GSM8K | 0.55 | 0.83 | 0.41 | 0.99 | 0.38 |
| TruthfulQA | 0.41 | 0.59 | 0.41 | 0.80 | 0.41 |
| MMLU | 0.07 | 0.07 | 0.09 | 0.07 | 0.14 |
| ARC | 0.02 | 0.02 | 0.03 | 0.02 | 0.04 |

Table 17: Contamination score returned by Shi (2023) on the contaminated portion of the benchmarks for PHI-3-SMALL. We compare openly malicious (OP) and evasively malicious (EV) actors.

|  | REF. | 1 Occ. | | 5 Occ. | |
| --- | --- | --- | --- | --- | --- |
|  |  | OP | EV | OP | EV |
| GSM8K | 0.30 | 0.65 | 0.39 | 0.68 | 0.39 |
| TruthfulQA | 0.18 | 0.28 | 0.21 | 0.39 | 0.22 |
| MMLU | 0.04 | 0.04 | 0.05 | 0.05 | 0.06 |
| ARC | 0.01 | 0.01 | 0.01 | 0.01 | 0.01 |

Table 18: Contamination score returned by Shi (2023) on the contaminated portion of the benchmarks for PHI-3.5-MINI. We compare openly malicious (OP) and evasively malicious (EV) actors.

|  | REF. | 1 Occ. | | 5 Occ. | |
| --- | --- | --- | --- | --- | --- |
|  |  | OP | EV | OP | EV |
| GSM8K | 0.63 | 0.85 | 0.74 | 0.87 | 0.76 |
| TruthfulQA | 0.28 | 0.37 | 0.33 | 0.41 | 0.37 |
| MMLU | 0.14 | 0.19 | 0.19 | 0.19 | 0.19 |
| ARC | 0.03 | 0.05 | 0.04 | 0.05 | 0.05 |

Table 19: Accuracy in % of LLAMA-3.1-8B on various benchmarks under contaminated and uncontaminated settings. C (resp. U) is measured on the contaminated (resp. uncontaminated) part of the test set. 2-sigma intervals are shown.

|  | REFERENCE | | 1 OCCURRENCE | | | | 5 OCCURRENCES | | | |
| --- | --- | --- | --- | --- | --- | --- | --- | --- | --- | --- |
|  |  | | OPEN | | EVASIVE | | OPEN | | EVASIVE | |
|  | C | U | C | U | C | U | C | U | C | U |
| GSM8k | $34.35_{\pm3.6}$ | $37.96_{\pm3.5}$ | $53.19_{\pm3.8}$ | $51.52_{\pm3.9}$ | $44.98_{\pm3.7}$ | $47.41_{\pm3.8}$ | $77.96_{\pm3.2}$ | $48.17_{\pm3.8}$ | $48.63_{\pm3.9}$ | $46.34_{\pm3.8}$ |
| MMLU | $62.88_{\pm4.3}$ | $59.27_{\pm4.2}$ | $82.56_{\pm3.5}$ | $59.27_{\pm4.4}$ | $64.71_{\pm4.2}$ | $60.29_{\pm4.5}$ | $96.96_{\pm1.5}$ | $62.53_{\pm4.2}$ | $64.50_{\pm4.3}$ | $59.67_{\pm4.4}$ |
| Arc | $75.64_{\pm3.6}$ | $71.13_{\pm3.8}$ | $92.62_{\pm2.1}$ | $77.32_{\pm3.5}$ | $78.90_{\pm3.5}$ | $74.91_{\pm3.5}$ | $99.49_{\pm0.6}$ | $77.84_{\pm3.3}$ | $79.76_{\pm3.1}$ | $74.40_{\pm3.5}$ |
| TruthfulQA | $47.42_{\pm4.8}$ | $46.67_{\pm4.8}$ | $77.89_{\pm3.9}$ | $66.17_{\pm4.6}$ | $58.23_{\pm4.6}$ | $51.11_{\pm4.9}$ | $96.31_{\pm1.8}$ | $64.69_{\pm4.7}$ | $63.14_{\pm4.9}$ | $53.33_{\pm4.8}$ |

Table 20: Accuracy in % of MISTRAL-7B on various benchmarks under contaminated and uncontaminated settings. C (resp. U) is measured on the contaminated (resp. uncontaminated) part of the test set. 2-sigma intervals are shown.

| | REFERENCE | | 1 OCCURRENCE | | | | 5 OCCURRENCES | | | |
| | | | OPEN | | EVASIVE | | OPEN | | EVASIVE | |
| | C | U | C | U | C | U | C | U | C | U |
| --- | --- | --- | --- | --- | --- | --- | --- | --- | --- | --- |
| GSM8k | $9.12_{\pm2.1}$ | $11.74_{\pm2.5}$ | $33.43_{\pm3.7}$ | $28.51_{\pm3.5}$ | $30.24_{\pm3.5}$ | $22.87_{\pm3.2}$ | $92.55_{\pm2.1}$ | $25.15_{\pm3.3}$ | $48.63_{\pm3.8}$ | $19.82_{\pm2.9}$ |
| MMLU | $50.10_{\pm4.4}$ | $43.88_{\pm4.4}$ | $81.74_{\pm3.3}$ | $48.68_{\pm4.3}$ | $51.93_{\pm4.6}$ | $46.84_{\pm4.4}$ | $96.55_{\pm1.6}$ | $42.36_{\pm4.4}$ | $52.54_{\pm4.4}$ | $44.60_{\pm4.1}$ |
| Arc | $61.92_{\pm3.9}$ | $58.76_{\pm4.0}$ | $91.08_{\pm2.3}$ | $66.49_{\pm3.8}$ | $70.67_{\pm3.6}$ | $62.03_{\pm3.9}$ | $99.49_{\pm0.6}$ | $59.45_{\pm4.0}$ | $68.95_{\pm3.7}$ | $57.90_{\pm4.0}$ |
| TruthfulQA | $43.98_{\pm4.6}$ | $46.17_{\pm4.8}$ | $79.36_{\pm3.9}$ | $59.26_{\pm4.9}$ | $64.86_{\pm4.8}$ | $56.05_{\pm4.7}$ | $93.12_{\pm2.4}$ | $59.01_{\pm4.6}$ | $55.77_{\pm4.9}$ | $46.67_{\pm4.9}$ |

Table 21: Accuracy in % of PHI-2 on various benchmarks under contaminated and uncontaminated settings. C (resp. U) is measured on the contaminated (resp. uncontaminated) part of the test set. 2-sigma intervals are shown.

| | REFERENCE | | 1 OCCURRENCE | | | | 5 OCCURRENCES | | | |
| | | | OPEN | | EVASIVE | | OPEN | | EVASIVE | |
| | C | U | C | U | C | U | C | U | C | U |
| --- | --- | --- | --- | --- | --- | --- | --- | --- | --- | --- |
| GSM8k | $25.23_{\pm3.2}$ | $24.24_{\pm3.4}$ | $47.11_{\pm3.8}$ | $39.48_{\pm3.9}$ | $36.78_{\pm3.6}$ | $35.21_{\pm3.7}$ | $60.33_{\pm3.8}$ | $39.48_{\pm3.8}$ | $46.05_{\pm3.9}$ | $35.37_{\pm3.7}$ |
| MMLU | $44.62_{\pm4.3}$ | $42.57_{\pm4.3}$ | $66.33_{\pm4.1}$ | $42.57_{\pm4.4}$ | $52.74_{\pm4.4}$ | $46.23_{\pm4.4}$ | $91.48_{\pm2.5}$ | $44.40_{\pm4.3}$ | $55.98_{\pm4.3}$ | $44.60_{\pm4.2}$ |
| Arc | $58.66_{\pm4.0}$ | $56.53_{\pm4.1}$ | $84.73_{\pm3.0}$ | $62.37_{\pm4.0}$ | $67.75_{\pm3.7}$ | $61.17_{\pm4.0}$ | $99.49_{\pm0.6}$ | $66.15_{\pm3.9}$ | $70.50_{\pm3.8}$ | $66.49_{\pm3.9}$ |
| TruthfulQA | $43.49_{\pm4.9}$ | $42.47_{\pm4.7}$ | $63.39_{\pm4.6}$ | $54.32_{\pm4.9}$ | $53.56_{\pm4.8}$ | $46.91_{\pm4.7}$ | $91.15_{\pm2.8}$ | $58.77_{\pm5.0}$ | $59.95_{\pm4.6}$ | $43.95_{\pm4.8}$ |

Table 22: Accuracy in % of PHI-3-SMALL on various benchmarks under contaminated and uncontaminated settings. C (resp. U) is measured on the contaminated (resp. uncontaminated) part of the test set. 2-sigma intervals are shown.

| | REFERENCE | | 1 OCCURRENCE | | | | 5 OCCURRENCES | | | |
| | | | OPEN | | EVASIVE | | OPEN | | EVASIVE | |
| | C | U | C | U | C | U | C | U | C | U |
| --- | --- | --- | --- | --- | --- | --- | --- | --- | --- | --- |
| GSM8k | $51.67_{\pm3.8}$ | $49.54_{\pm3.8}$ | $85.87_{\pm2.6}$ | $80.79_{\pm2.9}$ | $63.07_{\pm3.7}$ | $63.87_{\pm3.7}$ | $86.02_{\pm2.6}$ | $80.49_{\pm3.0}$ | $66.11_{\pm3.6}$ | $64.33_{\pm3.6}$ |
| MMLU | $65.92_{\pm4.2}$ | $58.66_{\pm4.4}$ | $83.77_{\pm3.3}$ | $72.51_{\pm4.0}$ | $67.95_{\pm4.0}$ | $59.67_{\pm4.3}$ | $95.54_{\pm1.8}$ | $74.13_{\pm4.0}$ | $71.60_{\pm4.1}$ | $66.19_{\pm4.2}$ |
| Arc | $71.70_{\pm3.5}$ | $71.65_{\pm3.6}$ | $92.28_{\pm2.2}$ | $88.14_{\pm2.7}$ | $89.37_{\pm2.5}$ | $87.97_{\pm2.8}$ | $45.28_{\pm3.8}$ | $25.95_{\pm3.5}$ | $87.48_{\pm2.6}$ | $87.97_{\pm2.7}$ |
| TruthfulQA | $57.00_{\pm5.2}$ | $56.05_{\pm4.8}$ | $60.20_{\pm4.6}$ | $52.10_{\pm5.0}$ | $50.86_{\pm5.0}$ | $49.63_{\pm4.9}$ | $84.52_{\pm3.6}$ | $68.40_{\pm4.5}$ | $61.92_{\pm4.8}$ | $54.57_{\pm4.9}$ |

Table 23: Accuracy in % of PHI-3.5-MINI on various benchmarks under contaminated and uncontaminated settings. C (resp. U) is measured on the contaminated (resp. uncontaminated) part of the test set. 2-sigma intervals are shown.

| | REFERENCE | | 1 OCCURRENCE | | | | 5 OCCURRENCES | | | |
| | | | OPEN | | EVASIVE | | OPEN | | EVASIVE | |
| | C | U | C | U | C | U | C | U | C | U |
| --- | --- | --- | --- | --- | --- | --- | --- | --- | --- | --- |
| GSM8k | $54.10_{\pm3.6}$ | $51.07_{\pm4.0}$ | $78.57_{\pm3.1}$ | $75.46_{\pm3.2}$ | $60.94_{\pm3.8}$ | $60.98_{\pm3.6}$ | $78.12_{\pm3.2}$ | $73.17_{\pm3.3}$ | $65.05_{\pm3.5}$ | $61.74_{\pm3.8}$ |
| MMLU | $50.51_{\pm4.2}$ | $47.86_{\pm4.3}$ | $75.05_{\pm3.9}$ | $66.80_{\pm4.2}$ | $71.60_{\pm3.9}$ | $66.60_{\pm4.1}$ | $89.66_{\pm2.6}$ | $67.41_{\pm4.3}$ | $71.40_{\pm4.2}$ | $66.60_{\pm4.3}$ |
| Arc | $54.37_{\pm3.9}$ | $54.81_{\pm3.9}$ | $88.34_{\pm2.6}$ | $82.99_{\pm3.1}$ | $84.56_{\pm2.9}$ | $84.19_{\pm2.9}$ | $95.54_{\pm1.7}$ | $85.05_{\pm3.1}$ | $85.25_{\pm2.9}$ | $82.65_{\pm3.2}$ |
| TruthfulQA | $60.69_{\pm4.8}$ | $60.00_{\pm4.6}$ | $50.37_{\pm4.8}$ | $46.67_{\pm4.9}$ | $51.35_{\pm4.6}$ | $46.17_{\pm4.7}$ | $66.83_{\pm4.8}$ | $55.80_{\pm5.1}$ | $58.23_{\pm4.7}$ | $51.85_{\pm4.9}$ |

Table 24: Table with assets used, description of their use and the license under which they are distributed. Sections are split by the type of asset: benchmarks, code repositories and then models.

| Asset | Description & Use | License Name |
|---|---|---|
| MMLU (Hendrycks et al., 2021) | Benchmark used for evaluation and contamination | MIT License |
| TruthfulQA (Lin et al., 2022) | Benchmark used for evaluation and contamination | Apache 2.0 License |
| GSM8k (Cobbe et al., 2021) | Benchmark used for evaluation and contamination | MIT License |
| ARC-Challenge (Clark et al., 2018) | Benchmark used for evaluation and contamination | CC-BY-SA-4.0 |
| OpenOrca (Lian et al., 2023) | Instruction-tuning dataset used in finetuning process | MIT License |
| (Shi, 2023) | Used repository to run the (Shi, 2023) baseline | Not Specified |
| MISTRAL-7B (Jiang et al., 2023) | Model finetuned to be contaminated | Apache 2.0 License |
| PHI-2 (Javaheripi et al., 2023) | Model finetuned to be contaminated | MIT License |
| PHI-3-SMALL (Abdin et al., 2024) | Model finetuned to be contaminated | MIT License |
| PHI-3.5-MINI (Abdin et al., 2024) | Model finetuned to be contaminated | MIT License |
| LLAMA-3.1-8B (Dubey et al., 2024) | Model finetuned to be contaminated | Llama 3.1 Community License Agreement |
| GPT-4 | Used to generate rephrased benchmarks | OpenAI Terms of Use |

