# OpenReview forum: "Evading Data Contamination Detection for Language Models is (too) Easy"
_ICLR.cc/2025/Conference — Submitted to ICLR 2025_

### Official Review · Reviewer_ujAG · 2024-10-26

**Soundness:** 2
**Presentation:** 1
**Contribution:** 2
**Rating:** 3
**Confidence:** 5

**Summary:**

This work aims to propose an evasive attack to bypass potential detection. The author first summarizes existing attacks, while studying potential detection strategies using metadata, reference models, semantic-preserving transformations, and different accessibility. Then, the author proposed that rephrasing can help to increase evasiveness.

**Strengths:**

+ This work studies a valid topic -- detecting more evasive attack to endanger LLM safety.
+ This work summarizes many attacks and detection efforts.

**Weaknesses:**

- Many settings are unclear: (1) For "contamination," there is no definition of the threat model regarding its knowledge, capabilities, or concrete examples in text-based attacks. (2) It is also unclear how the attacker influences benchmarks, what the adversarial samples look like, and what the adversarial objectives are. (3) Furthermore, defense settings are also unclear about the role of detector/defender and what actions can the defender take to eliminate poisoning? The author is suggested to add those information about both threat model and defender's strategies.

- The paper uses overly complicated wording, which creates barriers to understanding. Specifically: (1) "Contamination" is typically used as "poisoning." (2) The "actor" is generally referred to as "attacker" or "adversary." The "actor" typically links to specific roles (such as specific cyber threat patterns). (3) "De-Contamination" is rarely used. An alternative term is "mitigation." (4) "verbatim" could be "word-wise." (5) "Sample-level" could be more accurately expressed as "sentence (or word, token)-level." (6) It is inaccurate to use "Problematic Data Contamination" in Definition 3, as that definition is talking about possibility of injecting poisoning data D'.

- The author did not clearly explain why pre-training requires benchmarking data. The author probably means "public" or "crowdsourced" data.

- The paper over-claimed the contribution "We define four (de-)contamination settings..." where the settings are summarized, not defined in this paper.

- The role of model providers is unclear. In section 3, model providers are both attacker and defender. The author is suggested to distinguish model providers with adversaries (the "actor").

- The paper has poor structure and incomplete content: (1) There is no related work section. (2) The author did not detail the proposed rephrasing attack, such as what prompts or techniques are used for rephrasing. (3) Definition 4 simply repeats definition 1 and 2. (4) The Definition 5 should be highlighted with rigorous formulation with later technical design addressing it. (5) It would be better to introduce "TPR" (true positive rate) and "FPR" (false positive rate) in Figure 1.

- The contribution is not significant. The author simply uses "rephrasing," which lacks strong intuition and motivation for its usefulness.

- The experiments are simple, and the settings are problematic. Specifically: (1) the author mentions several detection strategies but does not provide corresponding experiments to address those detections. (2) The paper also fails to clarify how evasiveness is guaranteed -- Is it measured by lower TPR by detection methods? (3) No baseline attacks. (4) The author is suggested to discuss the trade-off between attack effectiveness and evasiveness.

- Many sentences seem GPT-generated. It is fine to use GPT to polish writing. However, the original writing maybe logically inconsistent and unclear, so GPT alone cannot produce better logic.

**Questions:**

Please see the comments above.

---

> ### Author Response · Authors · 2024-11-21
>
> We thank the reviewer for their detailed review and questions. We think there may have been a fundamental misunderstanding of our work’s setting which we address before answering the reviewer’s questions.
>
> **Setting of our Work**
> We do not study data poisoning, where an adversary/attacker (with limited power) aims to manipulate samples to reduce the performance of a model trained by a (different) defender. Rather, we study the (malicious) data contamination setting where the model provider/actor training the model can modify the training data. This model provider has full control over the training process. Hence, “contamination” in our work refers to samples or information from a benchmark being leveraged in training rather than adversarial manipulation of training samples.
>
> We believe most of the reviewer’s concerns arise from this misunderstanding, particularly regarding our wording, the role of model providers, the actions of a detection mechanism, and the objectives of the malicious actor. Therefore, the main part of our rebuttal (Q1-Q10) aims to resolve issues related to this misunderstanding, while the rest of the rebuttal aims to address the other concerns raised by the reviewer.
>
> **Q1. Why do the authors not use jargon from adversarial ML and in particular data poisoning, such as “threat model”, “defender”, and “attacker”?**
> We deliberately avoid such terminology to avoid confusion with the fundamentally different setting of data poisoning and its traditional attack-defense scenarios. These terms often imply adversarial roles and dynamics that do not align well with contamination evasion for the following reasons:
> - **Temporal and Role Reversal**: In classical attacks, adversaries act after defenses are implemented. In our work, the malicious model provider contaminates the model before any detection mechanisms are applied.
> - **Role Ambiguity**: Detection methods often use membership inference **attacks**, blurring the boundary between attacker and defender. These membership inference attacks require a “threat model” to list the assumptions under which they can effectively work. We present this “threat model” in the form of the detection methods assumptions in Section 4, but we chose not to refer to this as a threat model to avoid confusion about role ambiguity.
> - **No Classical Attack Equivalent**: There is no direct equivalent of our attack in the classical literature. As explained before, the most similar classical attack, data poisoning, is different from our setting.
> - **Model Providers are Neither Attackers nor Defenders**: Malicious model providers are adversaries in our attack, but proactive model providers can also perform contamination detection and therefore be a “defender”. Classifying model providers in this way would therefore cause confusion.
>
> **Q2. What actions can a defender take to avoid contamination?**
> Detection mechanisms cannot prevent contamination as the malicious actor controls the whole training process. The role of detection mechanisms is limited to detecting and reporting contamination.
>
> **Q3. How does an attacker influence benchmarks and what are their objectives?**
> The primary objective of a malicious model provider is to artificially inflate the model’s performance on a benchmark by contaminating the training data. Since the malicious actor has control over the entire training pipeline, they can influence training data however they want. Our attack can be broken down as follows:
> - **Rephrasing Benchmarks**: Using GPT-4 (details in Appendix E), benchmark samples are rephrased. This ensures the contaminated samples do not match verbatim benchmark entries, evading detection methods that rely on such matches.
> - **Expanding the Training Data**: These rephrased samples are added to the original training data, creating a contaminated dataset.
> - **Fine-tuning**: The model provider fine-tunes their model on the contaminated dataset.
>
> This approach demonstrates that even simple rephrasing can effectively evade detection while artificially inflating performance.
>
> **Q4. Are the four (de)-contamination settings merely summaries of prior work?**
> No, we are to the best of our knowledge the first to propose these distinctions in the context of contamination. This is highlighted by the fact that no current detection method considers evasively malicious actors. If the reviewer can point us to relevant literature, we will integrate it into our discussion.

---

> > ### Author Response · Authors · 2024-11-21
> >
> > **Q5. Can you clarify the wording used in your paper, e.g., verbatim, sample-level, and contamination?**
> > Yes, the wording can be clarified as follows:
> > - **Verbatim:** This is a widely used term that describes exact replication. The reviewer’s suggestion of "word-level" introduces ambiguity, as it is unclear whether it includes spaces, punctuation, or formatting.
> > - **Sample-level**: This refers to contamination affecting specific benchmark **samples**, which often consist of multiple sentences. Alternatives like "sentence-level" or "token-level" fail to capture the meaning of benchmark samples.
> > - **Contamination**: We use "contamination" instead of "poisoning" because the latter typically implies malicious intent, whereas contamination can also occur accidentally.
> >
> > **Q6. Can you clarify the structure of the paper?**
> > Please see Q1 in our main reply.
> >
> > **Q7. Can you clarify Definition 3?**
> > Please see Q2 of our main reply.
> >
> > **Q8. Why are there no baseline attacks?**
> > We are the first to propose an evasive contamination attack. As such, no directly comparable baseline attacks exist. However, comparing against openly malicious contamination strategies could serve as a pseudo-baseline, which we already include.
> >
> > **Q9. What is the difference between Definition 4 and Definition 1 and 2?**
> > Definition 4 categorizes model provider attitudes, while Definitions 1 and 2 focus on contamination types. To avoid confusion, we have renamed Definition 4 to “Actor Roles.”
> >
> > **Q10. Why does pretraining require benchmark data?**
> > Pretraining does not require benchmark data, and its inclusion constitutes contamination. We are not sure what comment the reviewer is referring to, but will rectify this if the reviewer can point out where it occurs.
> >
> > **Q11. Why did you discuss some detection methods, but did not include them in the experimental section?**
> > Please see Q3 of our main reply.
> >
> > **Q12. How is evasiveness guaranteed?**
> > We do not claim that our attack guarantees evasiveness. Instead, we demonstrate that current detection methods fail to reliably identify contamination caused by rephrased samples. This is shown by their performance in Section 6.3 and 6.4, which is indistinguishable from random guessing when contaminating with our attack. We indeed measure evasiveness as low TPR at fixed FPR of detection methods.
> >
> > **Q13. Can you discuss the tradeoff between effectiveness and evasiveness?**
> > As explained in Section 8, our attack sacrifices some effectiveness in favor of evasiveness. This is shown by the smaller performance gains observed compared to the openly malicious contamination strategy. The reduced effectiveness suggests that there may be alternative attack strategies that achieve a better balance, evading detection while maintaining stronger performance improvements.
> >
> > **Q14. How much did you use LLMs in your writing process?**
> > The paper was written by iteratively revising the text by the authors, with minimal use of LLMs. A few sentences were revised with the help of LLMs to enhance clarity, particularly in the appendix. Furthermore, all outputs of the models were carefully reviewed and considered to ensure correctness.
> >
> > Following the reviewer’s question, we analyzed the first few sections of our paper using GPTZero, a widely used AI-generated text detection tool. The results indicated negligible AI involvement (0%-5%). While these tools are not perfect, they are often better than human intuition.
> >
> > **Q15. Could you detail the prompts and techniques used for rephrasing?**
> > All the experimental details, including the prompts that were used to rephrase our data with GPT-4, the hyperparameters for fine-tuning, and how samples were given to the models, are given in Appendix E. If the reviewer thinks there is something missing in this appendix, we would be happy to add it.
> >
> > **Q16. Could you introduce TPR and FPR in Figure 1?**
> > Yes, we have now introduced these abbreviations in Figure 1.
> >
> > We hope that this addresses all the reviewers' questions and concerns. We are happy to discuss any further questions the reviewer might have.

---

### Official Review · Reviewer_MpuQ · 2024-10-29

**Soundness:** 3
**Presentation:** 2
**Contribution:** 3
**Rating:** 5
**Confidence:** 3

**Summary:**

The paper presents a examination of the vulnerabilities in current benchmarking practices for LLMs. They introduce Evasive Augmentation Learning (EAL), a technique that rephrases benchmark samples to boost performance on benchmarks and evade current detection methods.

**Strengths:**

- Method is simple and easy to understand.
- The topic is highly relevant to the current challenges in assessing the performance of large language models.

**Weaknesses:**

- Unclear definition on "Definition 3"
- The distinguish between openly malicious and evasively malicious is somewhat problematic. It's hard to find realistic usage when applying the open malicious scenario to any of the attackers. Openly malicious is easily getting rid of by the other detection method listed.
- Sections 4 and 5 would benefit from streamlining. There is some repetition and unnecessary content that could be condensed to improve clarity and conciseness.
- Although the paper states that code is provided, it does not include detailed descriptions of the processes for data rephrasing and contamination insertion, particularly regarding how rephrased data is generated.
- The authors overclaimed the contribution. Failure to consider the technical feasibility and applicability of updating baseline data in real-world applications, resulting in proposed attack methods that are more suitable for closed environments and may not work in open and real-time updating environments

**Questions:**

- Does the EAL technique circumvent all detection methods mentioned in A Taxonomy for Data Contamination in Large Language Models, including those that do not require pre-training data?

---

> ### Author Response · Authors · 2024-11-21
>
> We thank the reviewer for their detailed review and questions. We are pleased that they found our method simple and easy to understand, and the topic highly relevant. We address their questions and concerns in our response.
>
> **Can you clarify what you mean in Definition 3?**
> Please see Q2 of our main reply.
>
> **Why do you distinguish between openly malicious and evasive malicious actors?**
> We appreciate the reviewer's observation. The distinction between openly malicious and evasively malicious actors is included to highlight a critical point about the limitations of existing contamination detection methods. Many prior works evaluate their detection methods against scenarios where the contamination is obvious and straightforward—what we refer to as "openly malicious" actors. As the reviewer suggests, in real-world scenarios, a malicious actor is likely to use evasive strategies to avoid detection. Our purpose in making this distinction is to emphasize that evaluation should focus on evasively malicious actors, as they represent the realistic scenario. We will clarify this in the paper, emphasizing that openly malicious actors are discussed primarily to critique the inadequacy of existing evaluation practices (L168-174).
>
> **Could you streamline Sections 4 and 5 to remove some repetition and unnecessary content?**
> Section 4 discusses the assumptions current contamination detection methods make while Section 5 discusses how to evade these methods by systematically breaking their assumptions. We believe this helps provide an intuition for how we designed our evasion strategy and why it works. If the reviewer could clarify which sections they believe should be cut, we would be happy to edit them correspondingly.
>
> **Can you clarify some experimental details, specifically regarding data rephrasing?**
> All the experimental details, including the prompts that were used to rephrase our data with GPT-4, the hyperparameters for fine-tuning, and how samples were given to the models, is given in Appendix E. If the reviewer thinks there is something missing in this appendix, we would be happy to add it.
>
> **Does the EAL technique circumvent all detection methods mentioned in A Taxonomy for Data Contamination in Large Language Models, including those that do not require pre-training data?**
> Yes. We emphasize that our paper includes a superset of the contamination detection methods discussed in A Taxonomy for Data Contamination in Large Language Models. Please see Q3 for an analysis of why we evade all these attacks.
>
> **Could evaluators update the benchmark data in real-world applications to circumvent your attack?**
> Yes, as discussed in Section 7, dynamic benchmarks that are regularly updated could mitigate the risk posed by our method. However, static benchmarks are still the norm in most real-world scenarios, especially for widely recognized datasets. These benchmarks are critical because they directly influence public perception of model performance and are the primary targets for malicious actors seeking to inflate their results. While we acknowledge that dynamic benchmarks could reduce susceptibility to contamination, they are not yet widely adopted. Therefore, our focus on static benchmarks reflects the reality of current practices, which are most vulnerable to the described attack.
>
> We hope that this addresses all the reviewers' questions and concerns. We are happy to discuss any further questions the reviewer might have.

---

### Official Review · Reviewer_2ndD · 2024-11-03

**Soundness:** 3
**Presentation:** 3
**Contribution:** 3
**Rating:** 6
**Confidence:** 3

**Summary:**

This paper looks into the vulnerabilities and reliabilities of current data contamination detection methods. The authors categorize model providers based on their contamination practices, emphasizing malicious actors could intentionally contaminate models by rephrasing the benchmark data, thus evading detection while boosting model scores. Through experiments across several models and benchmarks, the authors show that the rephrasing technique could increase performance while bypassing existing detection methods.

**Strengths:**

- The paper introduces a novel categorization of model providers based on their data contamination practices.
- Provide a simple yet effective contamination technique to evade detection, highlighting limitations in current detection methods.
- The paper is well-organized, with each section building on the previous one. Key concepts like contamination types and detection vulnerabilities are presented accessibly, making the research easy to understand and follow.

**Weaknesses:**

The paper demonstrates the vulnerability of current detection methods to rephrasing attacks but does not explore potential defenses. Without suggestions for addressing this vulnerability, the work may feel incomplete.

**Questions:**

- Is the performance across different benchmarks consistent? Could you provide a more detailed breakdown of per-benchmark performance?
- The paper notes that the limited sample size of the contaminated set contributes to increased variance in results. Would it be feasible to increase the sample size of the contaminated dataset to reduce this variance and enhance result stability?

**Details Of Ethics Concerns:**

By highlighting potential methods to evade current data contamination detection methods, the authors seek to promote awareness and encourage the development of more secure and reliable evaluation frameworks. While the paper aims to strengthen detection methods, these findings could be misused if taken out of context. Therefore, an ethics review may be beneficial to ensure the findings are communicated and applied responsibly.

---

> ### Author Response · Authors · 2024-11-21
>
> We thank the reviewer for their detailed review and questions. We are pleased that they found our attack simple and effective, and the paper well-organized with each section building on the previous one. We will address their questions and concerns in our response.
>
> **Is the performance across different benchmarks consistent?**
> Yes, the performance gain is consistent. Detailed results for each model and benchmark are now provided in Tables 19–23, with a thorough discussion in Appendix F. While performance improvements vary across models and benchmarks, the contaminated models consistently outperform the baseline. The only exception is the TruthfulQA dataset for the Phi-3.5-Mini and Phi-3-Small models. We hypothesize that this anomaly may be due to an unfortunate loss spike in the fine-tuning process for this dataset.
>
> **Would it be feasible to increase the sample size of the contaminated dataset to reduce this variance and enhance result stability?**
> We are limited by the sizes of the benchmarks we use: TruthfulQA, GSM8k, and Arc-Challenge consist of only around 1000 samples each, which we all use. Only for MMLU could the contaminated dataset size be increased. However, our results are already statistically significant, with $2\sigma$ intervals of around 2%, compared to the 15% performance improvement achieved by our attack. Moreover, MMLU contributes to only one of the four datasets over which we average, so any reduction in variance from increasing its size would have only a small effect on our overall results.
>
> **Can you suggest defenses against your attack?**
> Generally, model contamination can be mitigated in two ways:  i) detecting contaminated models and excluding them from comparison and ii) evaluating models in a way that makes contamination ineffective at improving performance. As our attack evades all contamination detection methods we experimented with, we believe evaluation practices that make contamination ineffective (see Section 7) are the most promising.
>
> To suggest defenses against our attack, we can look at these alternatives: private benchmarks, human evaluation, and dynamic benchmarks. While private benchmarks and human evaluation provide an effective defense against our attack, they cannot be transferred to public benchmarks. In contrast, dynamic benchmarks could offer a defense. In these benchmarks, contamination is detected by comparing performance on data released before and after model release. If performance is significantly better on the data released before the model’s knowledge cutoff, contamination is likely. A similar approach could be applied to static, public benchmarks by generating more samples using the same protocol used when originally curating the benchmark, as has been done for, e.g., ImageNet (Recht et al. [1]).
>
> However, not only does extending datasets in this manner require considerable effort, but it also comes with further challenges. For example, the datasets should be kept private to avoid contaminating new models while still enabling private models, only available through API, to be evaluated.
>
> We hope that our work and other contamination evasion approaches lead to the development of more effective detection methods as has been the case in other fields where attacks and defenses are proposed separately (such as federated learning, adversarial machine learning, etc.).
>
> We hope that this addresses all the reviewer’s questions and concerns. We are happy to discuss any follow-up questions the reviewer might have.
>
> [1] Recht et al. "Do imagenet classifiers generalize to imagenet?." ICML 2019.

---

> > ### Comment · Reviewer_2ndD · 2024-12-03
> >
> > Most of my concerns have been resolved. I will keep my current rating.

---

### Official Review · Reviewer_uLUM · 2024-11-04

**Soundness:** 2
**Presentation:** 2
**Contribution:** 2
**Rating:** 3
**Confidence:** 4

**Summary:**

This paper focuses on the data contamination problem with an analysis of existing frameworks. The paper categorizes existing frameworks in 4 categories and reveals their weaknesses by a detailed study. The paper proposes a new attack strategy based on the analysis and observation.

**Strengths:**

1. The analysis of existing methods is reaonsable. The paper starts the discussion on categorizing existing paper with a figure and the categorization is reasonable.
2. The selection of base models and benchmarks is representative and comprehensive.

**Weaknesses:**

The paper lacks novelty and technical depth, and the presentation of the paper is hard-to-follow.
1. The paper mainly studies the existing methods, however, it lacks a detailed discussion on the proposed attack strategy and the reasonableness of such a strategy. As a reader, I cannot fully understand the main design of the strategy proposed after the analysis (In Sec 5.1 I suppose). It's hard to understand the method with a short paragraph and the paper lacks a formulation or figure for the strategy. 2. The experiments just focus on evaluating existing methods and it's hard to identify the section proving the effectiveness of the proposed strategy. The experimental design doesn't the support the claim of the paper and lacks crucial experimental results.
3. The organization of the paper is dispersive with no key idea. The presentation is not coherent. After discussing the experiment results, the paper switched to a discussion on section 7, but such a discussion is not related to the experiment results presented before. This section sounds like a "related work" section.

**Questions:**

Please refer to the weakness part about the paper structure, technical depth, and novelty.

---

> ### Author Response · Authors · 2024-11-21
>
> We thank the reviewer for their detailed review and questions. We will address their questions and concerns in our response.
>
> **Can you clarify the structure of the paper?**
> Please see Q1 in our main reply.
>
> **Can you explain the proposed strategy more thoroughly?**
> Our proposed attack strategy is straightforward: we rephrase benchmark data using GPT-4 and finetune our models on this rephrased data. This leads to contamination while evading existing detection methods. The rephrasing preserves the semantic meaning but changes the exact wording, effectively evading detection tools that rely on exact matches. Finetuning on this rephrased data significantly improves the model's benchmark performance without leaving traces of the original samples or their metadata.
>
> The prompts used for rephrasing are detailed in Appendix E. If there are specific aspects of the strategy you find unclear, we are happy to address them.
>
> **How does the experimental design support the claim of the paper?**
> The experimental design is structured to validate our key claim: that our attack evades existing contamination detection methods, while still improving model performance. This is demonstrated as follows:
> - **Section 6.2**: Shows performance improvements on standard benchmarks (Table 1), indicating successful contamination.
> - **Sections 6.3 and 6.4**: Demonstrate successful evasion of current detection methods (Tables 2, 3, and 4).
>
> We have excluded detection methods that are inherently unable to detect our attack due to assumptions that do not apply (e.g., requiring metadata to be leaked). Please see Q3 of our main reply for a thorough discussion of this exclusion. If there are specific experimental results you believe are missing, we welcome your feedback to strengthen the evaluation.
>
> **What are the technical contributions of the paper?**
> Our technical contributions are threefold:
> - A novel perspective on contamination, introducing a framework for categorizing potentially malicious model providers (Section 3).
> - A comprehensive analysis of existing detection methods, identifying their limitations (Section 4).
> - A new attack strategy that exploits these limitations and evades detection while achieving higher performance (Section 5)
>
> Thus, this work provides the first demonstration of a practical contamination method that systematically bypasses detection. We believe the simplicity of our attack is a strength rather than a weakness, making our findings even more significant.
>
> We hope that this addresses all the reviewer’s questions and concerns. We are happy to discuss any follow-up questions the reviewer might have.

---

### Author Response · Authors · 2024-11-21

$\newcommand{R}{\textcolor{green}{uLUM}}$
$\newcommand{S}{\textcolor{blue}{ujAG}}$
$\newcommand{T}{\textcolor{purple}{MpuQ}}$


We thank the reviewers for their detailed reviews and questions. We identified three common concerns, which we address below, before addressing individual questions in reviewer-specific responses.

**Q1. Can you clarify the structure of the paper? ($\R, \S$)**
The paper introduces a novel perspective on (detecting) data contamination by considering malicious model providers. The structure is designed to systematically build this perspective and design an attack that evades detection while significantly improving model performance:
- **Definition of contamination (Section 2)**: Provides an understanding of the problem and identifies the goal for a malicious model provider (problematic data contamination).
- **Framework for malicious model providers (Section 3)**: Categorizes behaviors of potentially malicious actors and reviews current model providers in this context.
- **Analysis of detection methods (Section 4)**: Investigates assumptions/requirements of existing detection methods in a new framework to reveal a key vulnerability.
- **Proposed attack strategy (Section 5)**: Proposes an attack/contamination strategy leveraging the vulnerability identified in Section 4.
- **Experimental validation (Section 6)**: Demonstrates the effectiveness of the attack and that it is indeed able to avoid detection while increasing performance.
- **Discussion and related work (Section 7)**: Discusses alternative evaluation strategies that prevent problematic contamination.

We have renamed the "Discussion" section to "Related Work" for clarity.

**Q2. Can you clarify Definition 3 on Problematic Data Contamination? ($\S, \T$)**
Definition 3 identifies the core objective of contamination: significantly improving benchmark performance. This involves comparing the performance of two models:
- One trained on uncontaminated data.
- One trained on the same uncontaminated data and on additional contaminated data (e.g., using rephrased or actual benchmark samples).

If the latter model significantly outperforms the former, we consider the contamination to be problematic. This definition provides a measurable and objective criterion for evaluating evasion attacks.

**Q3. Why did you not include all the detection methods you discussed in the experimental evaluation? ($\R,\S,\T$)**
To show how we evade all current detection methods, we analyze a large set of detection methods in Table 5 in Appendix B. Based on our analysis presented in Sections 4 and 5, we can exclude the following methods from this table for our evaluation:
- **Methods requiring oracle access**: Oracle-level access is not practical as training data is rarely published. However, we demonstrate in Appendix C that our evasion strategy is effective even under oracle-level detection methods.
- **Metadata-dependent detection methods**: Methods relying on metadata are inapplicable because we explicitly remove metadata from contaminated samples. This ensures that detection mechanisms dependent on metadata will fail on our approach.
- **Contamination detection approaches with no novel method**: Li (2023b), Xu et al. (2024), and Carlini et al. (2022) primarily offer analyses or conceptual discussions rather than detection strategies. As such, they do not introduce a new method that needs to be evaluated in our experiments.
- **Mathematically equivalent techniques**: Some works duplicate existing detection methods already included in our evaluation. For example, Watson et al. (2022) is a reimplementation of Mireshgallah et al. (2022), and Li (2023a) is functionally equivalent to Yeom et al. (2018).
- **Wrong-Option Contamination**: Similarly, Deng et al. (2023) require the inclusion of incorrect options of a benchmark in the training set, which we explicitly remove during the rephrasing process.
- **User-Level Contamination Methods**: Song et al. (2019) address user-level contamination in a privacy-based use case. They do not apply to the contamination scenario studied in this work.

After excluding the above methods, we evaluate our attack against the remaining approaches, including Zhu et al. (2023), Carlini et al. (2021), Miresgallah et al. (2022), Shi (2023), Shi et al (2023), Xie et al. (2024), Zhang and Wu (2024), Zhang et al. (2024c), and Yeom et al. (2018). Our results in Section 6 demonstrate that our attack effectively circumvents these methods by reducing their performance to random chance.

We included this discussion in Appendix B.

**Conclusion**
We hope to have addressed all the reviewers' concerns, remain happy to answer follow-up questions, and are looking forward to their replies and an active discussion phase.

---

### Author Response · Authors · 2024-11-29

We want to again thank all reviewers for their reviews. We believe to have addressed all of the reviewers’ concerns in our rebuttal. Given that the end of the discussion period is fast approaching, we would like to ask the reviewers to let us know about any points they feel are left unaddressed so that we have time to reply. We would further like to ask all reviewers to acknowledge that they read our response and to consider updating their review accordingly.

---

### Meta-Review · Area_Chair_pMMh · 2024-12-20

**Metareview:**

The authors study data contamination, with a particular focus on the adversarial setting, where language models ingest test sets, making benchmark comparisons invalid. The authors set up a categorization of different types of contamination (Fig 1), study existing membership inference and contamination detention methods, and argue that malicious contamination through an adversary is a serious and hard-to-detect problem.

Contamination of LLM benchmarks is a serious issue that erodes trust in evaluations, so this problem addresses an important problem, and the adversarial setting is generally under-studied relative to the careless-model-trainer setting.

However, the paper is in a bit of an odd spot in terms of contributions. Most of the paper is conceptual (setting up adversary vs not, in the first 5 pages or so) and past work has discussed some of these distinctions, though not necessarily as a formal taxonomy or threat model. It would be hard to say that this paper is the first to bring attention to the issue of adversarial contamination. The reviewers had some issues with writing, and I think this generally reflects the fact that it's hard to make sense of what these first few pages are really doing.

On the 'new method' front in section 5, the authors propose rephrasing as an attack, but Yang et al 2023 have made a rephrasing-based 'attack' for benchmark contamination. Of course, Yang et al do not evaluate against detection methods like Shi et al, but I don't think 'rephrasing as a contamination attack' is itself novel either.

I do think the paper has value as a benchmarking effort - showing that the rephrasing attack is effective at evading existing contamination detection methods - but I think this alone is hard to use as a reason to accept a paper, especially given recent works showing that MIA methods like Shi et al just don't work well even in the non-adversarial setting.

**Additional Comments On Reviewer Discussion:**

The authors clarified several misunderstandings by the reviewers, which was helpful for me in understanding what their underlying objections were and what the authors' contributions were.

The reviewers were less responsive than hoped for, but several of the points (novelty, some of the framing issues, etc) were carefully checked over by me in the decision process.

---

### Decision · Program_Chairs · 2025-01-22

Reject